# PHYSICS-INSPIRED RECONFIGURING MULTIMODAL LEARNING NETWORKS

## ABSTRACT

Despite recent progress, current multimodal fusion methods still face three practical issues: gradient interference between task and fusion objectives, fragility under missing modalities, and rigidity from enforcing uniform feature dimensions across modalities. We present Physics-Inspired Multimodal Reconfiguration (PMR), a Poisson–Nernst–Planck (PNP)–inspired structured prior for fusion. Drawing from the principles of conservation and single-potential-driven flow, PMR embeds these as (i) an information-preservation regularizer and (ii) a unified scalar potential that shapes gradient updates, mitigating interference between task and fusion objectives. This unified potential drives disentanglement of shared and modality-specific subspaces. A three-stage mapping (dissolution → dissociation → concentration) instantiates the prior to separate and reconstruct features, improving robustness to missing modalities and naturally supporting unequal feature dimensions. Across audio, image, video, and text, PMR consistently outperforms competitive baselines on classification and cross-modal retrieval, demonstrating the efficacy of a physics-inspired hybrid prior for multimodal learning.

## 1 INTRODUCTION

Multimodal learning integrates multiple distinct data modalities to improve model performance and generalization capabilities (Baltrušaitis et al., 2019), such as visual, auditory, and textual information. Cross-modal signals provide complementary views that enrich unimodal encoders and improve downstream accuracy (Lu et al., 2019; Zhao et al., 2024). However, jointly optimizing task and fusion objectives can introduce gradient interference, so an explicit fusion stage with its own objective is warranted to mediate and balance gradients (Kendall et al., 2018; Chen et al., 2018; Guo et al., 2024b). Therefore, introducing an additional fusion stage is necessary.

While an additional fusion stage is necessary, recent multimodal systems adopt specialized fusion structures and report strong numbers. Yet, under controlled stress tests (for example, models matched for capacity, reduced-label regimes, or compositionally novel splits), three recurring phenomena appear: (i) enforcing uniform feature dimensions across modalities can introduce redundancy and suppress modality-specific cues (Ying, 2019; Yao et al., 2024; Liu et al., 2021b); (ii) performance can degrade when inputs are incomplete (Ma et al., 2022; Wang et al., 2023a; Yao et al., 2024); and (iii) auxiliary fusion objectives can conflict with task objectives and cause gradient interference (Yang et al., 2023; Hazarika et al., 2020; Wang & Liu, 2020). High-capacity designs (e.g., the Graph Transformer in Meta-Transformer (Zhang et al., 2023b, Tab. 12)) make it difficult to separate gains from fusion design versus gains from capacity (Murata et al., 1994). These observations suggest that headline metrics alone are insufficient to identify the source of improvements.

To interpret these observations without presuming failure, we make explicit three working assumptions that often underlie current fusion practice: (i) uniform feature dimensions across modalities, (ii) complete modality availability, and (iii) auxiliary fusion objectives assumed to be unconditionally helpful, which can induce gradient interference with task objectives. Making these assumptions explicit and testable clarifies when simpler baselines or capacity scaling would suffice, and when fusion design itself contributes.

In contrast to these common assumptions, physics provides fundamental, invariant principles such as conservation laws and energy principles that can serve as structured priors for learning. These laws inherently guide system behavior towards stability and robustness. In light of this, we propose

PMR (Physics-inspired Multimodal Reconfiguration), which introduces a Poisson–Nernst–Planck-inspired prior (Granada & Kovtunenko, 2018). This prior draws from the concepts of **conservation (information preservation; no irreversible discard)** and **a single scalar potential (one energy surface; unified descent direction)**, and it is instantiated as an information preservation regularizer and a unified scalar objective that consolidates task and fusion terms. Specifically, our contributions include:

- Cast multimodal fusion as minimization of a scalar potential guided by entropy, combining task and preservation terms into one objective to provide a consistent target and explicit accounting of auxiliary effects.
- Embed a PNP-inspired prior by enforcing conservation via an information-preservation regularizer and unifying task and fusion into a single scalar objective, yielding one descent direction without irreversible discard.
- Propose PMR, a three-stage mapping reconfiguration with dissolution, dissociation, and concentration that instantiates it; in evaluations it acts as a simple strong fusion method and a validator under incomplete assumptions, yielding structured disentanglement, missing-modality robustness, and unequal-dimension support.

This paper casts standard multimodal tasks (classification and retrieval) into the PMR framework. Across public benchmarks spanning audio, image, video, and text, PMR delivers consistent, often substantial gains over strong baselines, supporting our central claim that a conservation-guided single-potential prior enhances transferability and improves fusion quality.

## 2 RELATED WORK

We review fusion-stage methods along three axes: Align. (whether feature dimensions are equalized before fusion), Ref. (whether mutual information related objectives or structures, such as contrastive or correlation, are used at the fusion stage), and Avail. (explicit support for missing modalities at inference). Supervision geometry differs by task: alignment tasks for matching or retrieval seek tight cross-modal coupling, whereas breadth tasks for classification prioritize coverage and robustness to heterogeneous or missing inputs. Tab. 1 summarizes representative fusion designs under these axes. For works that frame classification as pattern alignment (Chugh et al., 2020; Mittal et al., 2020; Cheng et al., 2023), we follow the settings reported in the original papers. Perceiver does not report multimodal correlation results.

Table 1: Fusion-stage comparison. Align.: dimensional alignment required. Ref.: mutual-information–related objective/structure at fusion. Avail.: explicit design for missing modalities.

| Method | Align. | Ref. | Avail. | Modality & Related Task |
|---|---|---|---|---|
| CMMP (Radford et al., 2021) | ✓ | ✓ | – | I and V, retrieval |
| METER (Dou et al., 2021) | ✓ | ✓ | – | I and T, matching |
| APIVR (Xu et al., 2020) | ✓ | ✓ | – | I and V, retrieval |
| MAP-IVR (Liu et al., 2021a) | × | ✓ | – | I and V, retrieval |
| DI-VTR (Guo et al., 2024a) | ✓ | ✓ | – | I and V, retrieval |
| AADML (Zeng et al., 2024) | ✓ | ✓ | – | I and V, retrieval |
| CMKT (Zhou et al., 2025) | ✓ | ✓ | – | I and V, retrieval |
| EquiAV (Kim et al., 2024) | ✓ | × | – | A and V, retrieval |
| MDS (Chugh et al., 2020) | ✓ | ✓ | – | A and I, matching or classification |
| Emo-Foren (Mittal et al., 2020) | ✓ | ✓ | – | A and V, matching or classification |
| VFD (Cheng et al., 2023) | ✓ | ✓ | – | A and I, matching or classification |
| MISA (Hazarika et al., 2020) | ✓ | ✓ | ✓ | A V T, emotion classification |
| UAVM (Gong et al., 2022) | ✓ | × | ✓ | A and V, event matching |
| AVoiD-DF (Yang et al., 2023) | ✓ | ✓ | × | A and V, deepfake classification |
| DrFuse (Yao et al., 2024) | ✓ | ✓ | ✓ | EHR and CXR, classification |
| QMF (Zhang et al., 2023a) | × | ✓ | ✓ | I and D; I and T, classification |
| MLA (Zhang et al., 2024) | ✓ | ✓ | ✓ | A and V; I and T; A I T, classification |
| MDF-FND (Lv et al., 2025) | ✓ | ✓ | ✓ | I and T, classification |
| MBT (Nagrani et al., 2021) | ✓ | × | × | A and V, event classification |
| Perceiver (Jaegle et al., 2021) | ✓ | × | × | generic |

## 3 PRELIMINARY

This section serves as a preparatory foundation, introducing the symbolic system, the optimization objectives from an entropic perspective, and the underlying physical background.

Consider inputs with $M$ modalities, where $m \in \{1, 2, \ldots, M\}$ represents different modalities. A dataset containing $N$ samples is examined. Let the inputs be $X = \{X_1, X_2, \ldots, X_N\}$, where each specific sample $i \in \{1, 2, \ldots, N\}$ is represented as $X_i = \{X_i^{(1)}, X_i^{(2)}, \ldots, X_i^{(M)}\}$. The outputs are $Y = \{Y_1, Y_2, \ldots, Y_N\}$, where each pair $\{X_i, Y_i\}$ forms a sample pair. $X_i^{(m)}$ denotes the original sample of modality $m$, which has different shapes, while the shape of $Y_i$ depends on the specific dataset and downstream task. For each modality $m$, a specific feature extractor $f^{(m)}(\cdot, \theta^{(m)})$ with parameters $\theta^{(m)}$ is employed for feature extraction. The feature generated from feature extractor $f^{(m)}(X_i^{(m)}, \theta^{(m)})$ is denoted as $V_i^{(m)}$. The fused feature representation capturing the multimodal interactions of sample $i$ is denoted as $Z_i = \{Z_i^{(1)}, Z_i^{(2)}, \cdots, Z_i^{(M)}\}$. A concatenated column vector

$$V = f(X, \theta) = (f^{(1)}(X^{(1)}, \theta^{(1)}), f^{(2)}(X^{(2)}, \theta^{(2)}), \cdots, f^{(M)}(X^{(M)}, \theta^{(M)}) \in \mathbb{R}^{Ml},$$

represents the global features, where $\theta = \{\theta^{(1)}, \theta^{(2)}, \ldots, \theta^{(M)}\}$.

### 3.1 MULTIMODAL TASK OBJECTIVES VIA ENTROPY PERSPECTIVE

Cross-modal evidence should sharpen within-modal discrimination while keeping cross-modal noise bounded (Guo et al., 2024b; Zhao et al., 2024). We treat fusion as a router at a shared bottleneck, not a generator: it rescales one task gradient and routes it to the modality-specific encoders. With information conserved, fusion cannot create information; it can only reorder and filter. Heavy operations on task-irrelevant content at fusion induce brittleness under missing modalities and weaken transfer (Du et al., 2023; Wang & Liu, 2020; Boudiaf et al., 2020). We therefore shift the burden to the private encoders: fusion must not diminish the mutual information between any modality and the label. Encoders perform selection and compression (Kawaguchi et al., 2023), while fusion confines itself to reordering and disentanglement, suppressing spurious cross-modal dependencies.

Formally, let $f(X, \theta) = \{f^{(m)}(X^{(m)}, \theta^{(m)})\}_{m=1}^M$ be modality-specific extractors, $g(\cdot, \theta^F)$ the fusion module, and $h(\cdot, \theta^C)$ the task head. We minimize the task conditional entropy while constraining each extractor to remain locally optimal for its unimodal conditional entropy (Jiang et al., 2023):

$$\min_{\theta, \theta^F, \theta^C} H\big(Y \mid h\big(g[f(X, \theta), \theta^F], \theta^C\big)\big)$$

$$\text{s.t.} \quad \forall m \in \{1, \ldots, M\}, \ \ \theta^{(m)} \in \arg\min_{\hat{\theta}} H\Big(Y \,\Big|\, f^{(m)}\Big(X^{(m)}, \hat{\theta}\Big)\Big). \tag{1}$$

Intuitively, the constraint preserves task-relevant information at the extractor level (preventing fusion from degrading modality-specific predictiveness), while the global objective aligns the fused representation to the downstream task. In practice we enforce the constraint via a soft penalty (i.e., a single scalar potential added to the task loss); full derivations, the two-stage decomposition, and mutual-information discussion are deferred to the appendix C.

### 3.2 POISSON–NERNST–PLANCK (PNP) BACKGROUND AND RATIONALE

In many natural media, structure emerges only when a field does work against disorder. Without driving, diffusion raises entropy and mixes species (Zemansky & Menger, 1952); with a field, a single scalar potential $\phi$ encodes preference and its gradient $-\nabla\phi$ induces directed drift. A continuity law closes the description by enforcing that total amount is accounted for: redistribution may occur, but there is no creation or annihilation. This pair (one potential, one conservation rule) is a minimal, coordinate-free template that avoids ad-hoc mechanisms while guaranteeing a coherent direction of evolution. In standard settings (Granada & Kovtunenko, 2018), PNP-type systems admit a free-energy dissipation law (the energy decreases in time under no-flux boundaries), which provides a Lyapunov view of the dynamics (Kinderlehrer et al., 2017). PNP couples conservation, drift–diffusion transport, and field–charge coupling (Schuss et al., 2001):

$$\partial_t c_i + \nabla \cdot \mathbf{J}_i = 0, \qquad \mathbf{J}_i = -D_i \nabla c_i + \frac{D_i z_i e}{k_B T} c_i \mathbf{E}, \qquad -\nabla \cdot (\varepsilon \nabla \phi_{\text{int}}) = \sum_i z_i e c_i, \tag{2}$$

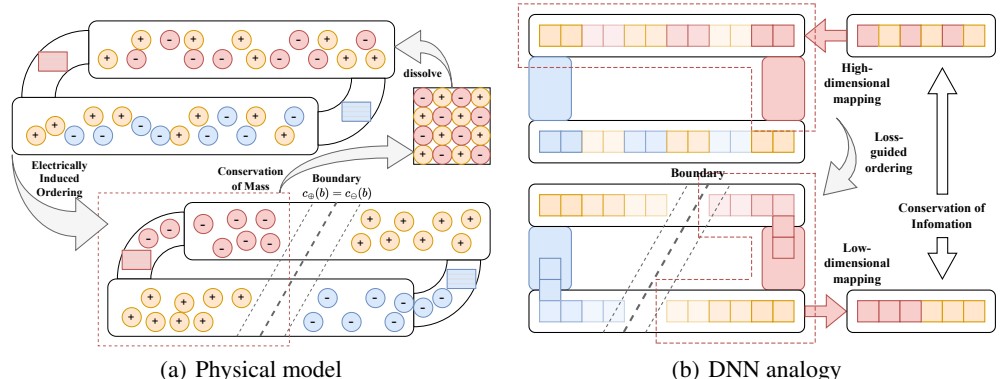

|                  |                  |
| :--------------: | :--------------: |
| (a) Physical model | (b) DNN analogy |

Figure 1: Cyclic electrolysis diagram. Curved pipelines represent ideal salt bridges, with cross-modal loss stimulation from both ends. The bold dashed line denotes the actual $b$ of $c_\oplus(b) = c_\ominus(b)$., and the finer dashed line shows the boundary interval where Eq. (3) holds.

where $c_i$ is the concentration of species $i$, $\mathbf{J}_i$ its flux, $D_i$ a diffusion coefficient, $z_i e$ the charge, $\mathbf{E} = -\nabla\phi_{\text{int}}$ the internal field, and $\varepsilon$ the permittivity. An external potential $\phi_{\text{ext}}$ can be superposed so that the total potential is $\phi = \phi_{\text{int}} + \phi_{\text{ext}}$. The model assumes only transport plus a field and a conservation law, and therefore appears across diverse media.

A steady state satisfies $\partial_t c_i = 0$ and $\nabla \cdot \mathbf{J}_i = 0$. In a one-dimensional domain of length $l$, drift in $\nabla\phi$ competes with diffusion. Under standard boundary conditions there exists a separation interface location $b \in [0, l]$ at which opposite species balance (for two species, $c_\oplus(b) \approx c_\ominus(b)$), and the net flux vanishes. Increasing the effective length enlarges the region where drift dominates diffusion, sharpening separation; when the domain greatly exceeds the screening length, profiles approach step-like distributions with unlike species clustering toward different ends. The pair $(b, l)$ thus summarizes where separation occurs and how sharply it is expressed. At equilibrium, as shown in Fig. 1(a), critical potentials $\phi_c$ and cell length $l_c$ ($l_c > \lambda_D$) satisfy

$$\left[ \frac{\partial\phi}{\partial x}\Big|_{x=b} - \frac{\partial\phi}{\partial x}\Big|_{x=0} \right] \approx -\frac{e}{\varepsilon_0} \int_0^b z_\ominus\, c_\ominus(x, t)\, dx. \tag{3}$$

A single potential yields one consistent driving direction (drift along $-\nabla\phi$) rather than multiple unrelated drives; a continuity law supplies the budget constraint that prevents trivial collapse by depletion. Together they form a non-ad-hoc, geometry-agnostic description in which directed rearrangement coexists with total-amount accounting, matching the qualitative requirements of any process that must reorganize content without destroying it. The input content is independent of the description of the output results and the process. We conducted derivations and simple numerical simulations in Appendix G, and an analogy between loss and $\phi_{\text{ext}}$ in Appendix F.

## 4 METHOD

This section shifts the perspective to DNNs and introduces the structure of Physics-inspired Multimodal Reconfiguration Network. The comparison of terms is shown in Tab. 2. $n, b$ is hyper-parameters, and in our default setting, $b^{(m)} = b$.

Table 2: 1D PNP versus the proposed PMR.

| 1D PNP | PMR |
| :--- | :--- |
| Positive/negative species | Shared/specific feature components |
| Charge quantity $(z_\oplus, z_\ominus)$ | Feature ratio and learned splitting at $b$ |
| External potential $\phi_{\text{ext}}$ | Total objective $\Phi = L_{\text{task}} + \lambda L_{\text{preserve}}$ |
| Drift–diffusion transport | Gradient descent on $\Phi$ |
| Dissolution (spreading) | High-dimensional mapping (4) |
| Concentration (gathering) | Low-dimensional mappings (5) |
| Mass conservation | Information-preservation regularizer (6) |
| Interface location | Dissociation location (boundary) $b \in [0, l]$ |
| Cell length $l$ | Magnification factor $n$ (effective length) |

**PNP provides two key principles**:
a scalar potential and a continuity law, which we discretize and use as a structured prior for multimodal fusion without solving the PDEs. **The scalar potential** enforces a unified objective at the shared bottleneck, reducing multiobjective interference by guiding updates in one direction. **The continuity law** ensures that fusion reorders rather than destroys information, preserving recoverable data.

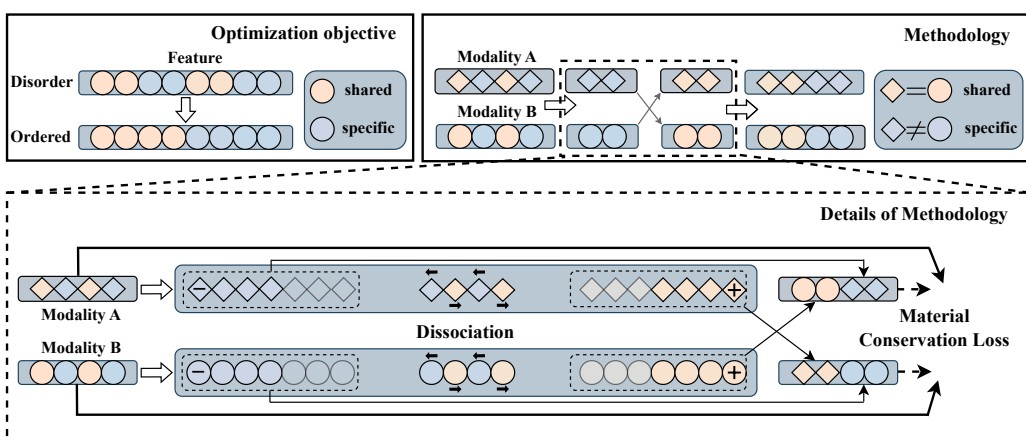

Figure 2: The schematic diagram of the PMR process. Ideally, the shared component is completely consistent, while the modality-specific component is entirely inconsistent. PMR guides the feature separation of a specific modality through the shared features of other modalities. We carried out a more detailed conceptual transfer and supplemented in Appendix E.4.

Together, these principles couple discriminativeness with recoverability, reduce training conflicts, and enhance robustness to missing modalities and distribution shifts, while allowing the task head to focus on relevant components: encoders extract, and the head selects.

Let $V_i^{(m)} \in \mathbb{R}^{l^{(m)}}$ be the extractor output for modality $m$, and let $l^* = \min_m l^{(m)}$, $\mathbf{P}$ is a mapping network (simple linear, or a structure) that maps vectors to a new dimension. Reconfiguration proceeds in three steps with shared notation across modalities:

$$\theta^{\mathrm{PMR}} = \left\{ \left( \mathbf{P}_{\mathrm{dis}}^{(m)}, \mathbf{P}_{\mathrm{shared}}^{(m)}, \mathbf{P}_{\mathrm{specific}}^{(m)}, \mathbf{P}_{\mathrm{recon}}^{(m)} \right) \mid m = 1, \ldots, M \right\}.$$

Dissolve (high-dimensional spreading). Each $V_i^{(m)}$ is mapped to $n\, l^{(m)}$ and split at a boundary $b^{(m)}$:

$$\hat{Z}_i^{(m)} = \mathbf{P}_{\mathrm{dis}}^{(m)} V_i^{(m)}, \quad \hat{Z}_{i,\mathrm{shared}}^{(m)} \in \mathbb{R}^{b^{(m)}}, \ \hat{Z}_{i,\mathrm{specific}}^{(m)} \in \mathbb{R}^{nl^{(m)}-b^{(m)}}. \tag{4}$$

Dissociate–concentrate (low-dimensional gathering). Let $k = (m+1)\%M$ denotes the next modality of $m$, Use $\hat{Z}_{i,\mathrm{shared}}^{(k)}$ from $k$ instead of $m$, shared and specific parts are projected to target sizes:

$$Z_{i,\mathrm{shared}}^{(m)} = \mathbf{P}_{\mathrm{shared}}^{(m)} \hat{Z}_{i,\mathrm{shared}}^{(k)} \in \mathbb{R}^{l^*}, \qquad Z_{i,\mathrm{specific}}^{(m)} = \mathbf{P}_{\mathrm{specific}}^{(m)} \hat{Z}_{i,\mathrm{specific}}^{(m)} \in \mathbb{R}^{l^{(m)}}. \tag{5}$$

The reconfigured output for modality $m$ concatenates its modality-specific part with the shared part supplied by the next modality, yielding $Z_i^{(m)} \in \mathbb{R}^{l^{(m)}+l^*}$. This also serves as the fusion output.

Preservation (continuity prior). A reconstruction operator maps back to the original feature space and enforces information preservation; two instantiations are used:

$$\mathcal{L}_{\mathrm{sPMR}} = \sum_{m=1}^{M} \left\| V_i^{(m)} - \mathbf{P}_{\mathrm{recon}}^{(m)} Z_i^{(m)} \right\|_2^2, \qquad \mathcal{L}_{\mathrm{rPMR}} = \sum_{m=1}^{M} \left( 1 - \cos\left( V_i^{(m)}, \mathbf{P}_{\mathrm{recon}}^{(m)} Z_i^{(m)} \right) \right). \tag{6}$$

The training objective is $\Phi = L_{\mathrm{task}} + \lambda L_{\mathrm{preserve}}$, where $L_{\mathrm{preserve}} \in \mathcal{L}_{\mathrm{sPMR}}, \mathcal{L}_{\mathrm{rPMR}}$. The scalar potential perspective ensures optimization follows a single descent direction, with the preservation term acting as a budget to prevent irreversible deletion. **In brief (See Appendix D), scalar-potential methods are task-aligned and composable, while non-scalar-potential methods are task-agnostic, leading to gradient interference, path dependence, and reduced robustness.**

The above design addresses the three incomplete assumptions from Sec. 1 as follows: (i) by optimizing a single total objective on the same reconfiguration operators, gradient interference between task and fusion is turned into an explicit one-step trade-off rather than a stage-wise or module-wise tug-of-war; (ii) by enforcing preservation, missing-modality cases lead to redistribution instead of collapse, since reconfiguration is constrained to be rearrangement rather than destruction; (iii) by dissolving to $n\, l^{(m)}$ and concentrating to $(l^{(m)}, l^*)$, the method avoids rigid uniform dimensions and performs separation at an enlarged effective length before projecting to the required sizes.

## 5  EXPERIMENTS

This section validates that the incomplete assumption discussed in Sec. 1 is widely prevalent among related approaches, and evaluates the performance of PMR across multiple tasks. Due to space limitations, PMR variant, implementation details are provided in the Appendix A.

### 5.1  TASKS AND BASELINES

We evaluate PMR on four datasets spanning retrieval, detection, and classification: ActivityNet (Heilbron et al., 2015) for image–video retrieval, VGGSound (Chen et al., 2020) for audio–video retrieval, FakeAVCeleb (Khalid et al., 2021) for deepfake detection with cross-modal label inconsistency, and Food-101 (Wang et al., 2015) for classification with aligned labels. We consider three training regimes to separate fusion behavior from representation learning effects: fixed feature extractors (to isolate fusion/entropy effects), end-to-end training (to assess the impact on backbone gradients), and a flexible setting that uses each method's native extractor when available.

Metrics follow standard practice: retrieval on ActivityNet and VGGSound uses mAP@$n$ (selecting the target from $n$ candidates); Food-101 report accuracy (%), FakeAVCeleb reports AUC additionally due to class imbalance. Baselines mirror task families: for classification we use Concat with a task head; for retrieval we use CMMP (a CLIP-style shared-feature baseline) and optionally Perceiver, MoE, SRHT. PMR has two variants, strict and relaxed (sPMR with $\mathcal{L}_{sPMR}$/rPMR with $\mathcal{L}_{rPMR}$); when composed with another method we write sG-/rG- (e.g., sG-CMMP, rG-SRHT, sG-UMoE).

### 5.2  RETRIEVAL

This subsection focuses on retrieval, probing the necessity of native unequal-length support and the effect of task-agnostic gradients. We compare disentanglement-based and metric-learning heads under varying levels of label noise and sample noise. To avoid confounding from loss-scale discrepancies, we adopt Xu et al. (2020)'s contrastive loss formulation for contrastive methods.

#### 5.2.1  IMAGE–VIDEO RETRIEVAL WITH FLEXIBLE UNEQUAL-LENGTH FUSION

**Setup.** We evaluate image–video retrieval on ActivityNet under two input configurations: unequal-length (128d-4096d, 128d image, 4096d video, following Xu et al. (2020)) and equal-length (128d-128d, mapping video feature to 128d additionally). Baselines include alignment method: CMMP (Radford et al., 2021), METER (Dou et al., 2021); disentanglement method (shared only): APIVR (Xu et al., 2020), MAP-IVR (Liu et al., 2021a), DrFuse (Yao et al., 2024), MISA (Hazarika et al., 2020); and structure-focused methods: Perceiver (Jaegle et al., 2021), DI-VTR (Guo et al., 2024a), and guided metric learning based: AADML (Zeng et al., 2024), CMKT (Zhou et al., 2025). Among these methods, UAVM and Perceiver rely solely on label-guided supervision; CMMP, AADML, DI-VTR, CMKT, and PMR use purely contrastive learning; whereas METER, MISA, DrFuse, MAP-IVR, and APIVR combine both paradigms. We report mAP@10/50/100.

**Results.** As shown in Tab. 3, label-guided methods rely on CE anchors; variation stems from the pre-training target and the strength of cross-modal geometric constraints. Pure SSL/contrastive methods center on pairwise similarity; splits are driven by hard-negative mining and structural priors—unified objectives with explicit geometry reshaping are most stable. Dual-track (CE+contrastive) methods gain in-domain but risk CE–contrastive gradient tension, dictated by alignment/disentanglement design. The unequal-length regime preserves high-capacity video features, raising the ceiling yet increasing mismatch and thus rewarding structured priors. PMR leads under both regimes via a single-potential objective and an information-preservation prior; rG slightly exceeds sG due to more flexible composition and easier optimization, especially with unequal-length inputs.

#### 5.2.2  AUDIO–VIDEO RETRIEVAL WITH NOISY LABEL AND SCALING

*Setup.* We evaluate audio–video retrieval on VGGSound (Chen et al., 2020) with a UAVM-style pipeline: ResNet-34 (He et al., 2016) encoders (512-D) are frozen, no temporal modeling is used, and fusion heads take $\mathbb{R}^{B \times C}$ inputs with the same contrastive objective as in ActivityNet. Baselines are grouped into three families consistent with ActivityNet: (i) alignment heads (UAVM (Gong et al., 2022), EquiAV); (ii) disentanglement/shared-only heads (MAP-IVR (Liu et al., 2021a)),

Table 3: Comparison on ActivityNet (I–V retrieval). We report mAP@10/50/100.

| | Method | mAP@10 | mAP@50 | mAP@100 |
|---|---|---|---|---|
| 128-128 | CMMP (Radford et al., 2021) | 0.219 | 0.195 | 0.189 |
| | Perceiver (Jaegle et al., 2021) | 0.271 | 0.253 | 0.240 |
| | METER (Dou et al., 2021) | 0.254 | 0.231 | 0.223 |
| | MISA (Hazarika et al., 2020) | 0.263 | 0.238 | 0.229 |
| | MAP-IVR (Liu et al., 2021a) | 0.341 | 0.306 | 0.294 |
| | UAVM (Gong et al., 2022) | 0.340 | 0.308 | 0.297 |
| | DrFuse (Yao et al., 2024) | 0.345 | 0.317 | 0.305 |
| | AADML (Zeng et al., 2024) | 0.338 | 0.306 | 0.296 |
| | DI-VTR (Guo et al., 2024a) | 0.347 | 0.320 | 0.306 |
| | CMKT (Zhou et al., 2025) | 0.335 | 0.304 | 0.294 |
| | **sG-CMMP** | 0.349 | 0.323 | 0.308 |
| | **rG-CMMP** | 0.351 | 0.324 | 0.310 |
| 4096-128 | APIVR (Xu et al., 2020) | 0.264 | 0.249 | 0.232 |
| | MAP-IVR (Liu et al., 2021a) | 0.349 | 0.322 | 0.311 |
| | MISA (Hazarika et al., 2020) | 0.268 | 0.244 | 0.236 |
| | DrFuse (Yao et al., 2024) | 0.352 | 0.325 | 0.313 |
| | **sG-CMMP** | 0.355 | 0.327 | 0.315 |
| | **rG-CMMP** | 0.356 | 0.330 | 0.316 |

Table 4: Audio–Video retrieval on VG-GSound. We report R@5 / R@10.

| | Method | R@5 | R@10 |
|---|---|---|---|
| A→V | MAP-IVR (Liu et al., 2021a) | 0.313 | 0.372 |
| | UAVM (Gong et al., 2022) | 0.365 | 0.453 |
| | EquiAV (Kim et al., 2024) | 0.342 | 0.420 |
| | AADML (Zeng et al., 2024) | 0.331 | 0.401 |
| | CMKT (Zhou et al., 2025) | 0.352 | 0.437 |
| | DrFuse (Yao et al., 2024) | 0.370 | 0.460 |
| | **rG-SRHT** | 0.353 | 0.432 |
| | **rG-CMMP** | 0.378 | 0.469 |
| | **rG-UMoE** | 0.395 | 0.482 |
| V→A | MAP-IVR (Liu et al., 2021a) | 0.297 | 0.356 |
| | UAVM (Gong et al., 2022) | 0.365 | 0.453 |
| | EquiAV (Kim et al., 2024) | 0.335 | 0.408 |
| | AADML (Zeng et al., 2024) | 0.324 | 0.395 |
| | CMKT (Zhou et al., 2025) | 0.345 | 0.430 |
| | DrFuse (Yao et al., 2024) | 0.362 | 0.450 |
| | **rG-SRHT** | 0.350 | 0.435 |
| | **rG-CMMP** | 0.375 | 0.466 |
| | **rG-UMoE** | 0.391 | 0.478 |

DrFuse (Yao et al., 2024); and (iii) structure- or label-guided metric heads (CMKT (Zhou et al., 2025), AADML (Zeng et al., 2024), with AADML as a label-guided distillation head). VGGSound has noisy labels and imperfect A–V alignment, so label-guided heads may inherit anchor noise whereas purely contrastive heads are more robust but susceptible to hard-negative collisions. We report R@5/R@10 for A→V and V→A. PMR heads include rG-SRHT, rG-CMMP, and rG-UMoE, where rG-UMoE applies a sparse MoE after the shared output to enable expert scaling/discard, rG-SRHT performs dimensional operations via the Subsampled Randomized Hadamard Transform (SRHT), requiring no learnable parameters and no explicit assignment step.

*Results.* Results are shown in Tab. 4. Under noisy alignment, PMR heads outperform the six baselines in both directions. Alignment heads degrade with anchor noise; shared-only heads curb channel bias but cap the ceiling; structure-/label-guided heads improve in-domain alignment yet are noise-sensitive. Directional asymmetry (A→V > V→A) reflects higher ambiguity of audio queries, while PMR's single-potential objective and information-preservation prior reduce gradient conflicts and modality cannibalization. Despite having no trainable parameters, PMR surpasses several parameterized baselines. When instantiated as rG-CMMP with a lightweight FFN-style head, it achieves state-of-the-art performance; extending this head to a sparse Mixture-of-Experts yields further gains.

## 5.3 CLASSIFICATION

This section assesses the classification task, specifically focusing on three axes of practical significance: (i) robustness to missing modalities, (ii) sensitivity to variable-length inputs and temporal alignment, and (iii) the scalability of fusion techniques from single-modal to tri-modal settings. The findings suggest that task-agnostic fusion is not always optimal, and that light task-specific adaptation may be necessary for achieving the best performance across varying tasks and datasets.

### 5.3.1 FAKEAVCELEB: RESILIENCE TO MISSING MODALITIES AND GRADIENT STABILITY

*Setup.* We evaluate deepfake detection on FakeAVCeleb (Khalid et al., 2021), a highly imbalanced benchmark (overall real:fake ≈ 1:39; video 1:19; audio 1:1). To ensure fair comparison, we adopt the feature extractors used in the original methods: MISA (Hazarika et al., 2020) with sLSTM (Hochreiter & Schmidhuber, 1997), UAVM (Gong et al., 2022) with ConvNeXT-B (Todi et al., 2023), and R(2+1)D-34 (Tran et al., 2018) for DrFuse (Yao et al., 2024), Perceiver (Jaegle et al., 2021), our concat baseline, and PMR backbones. For AVoiD-DF (Yang et al., 2023), VFD (Cheng et al., 2023), Emo-Foren (Mittal et al., 2020), and MDS (Chugh et al., 2020), we report the results published in Yang et al. (2023) under the same split. 'Unimodal' represents end-to-end single-modal training, 'Unimodal lin.' refers to using the output of the feature extractor from the multimodal network

Table 5: Performance on the FakeAVCeleb dataset. 'A' and 'V' indicate using audio and video modalities respectively, 'AV' represents complete samples.

| Method | ACC (%) | | | AUC (%) | | |
|---|---|---|---|---|---|---|
| | A | V | AV | A | V | AV |
| Unimodal (Tran et al., 2018) | 99.68 | 95.35 | - | 99.65 | 50.31 | - |
| Concat (Tran et al., 2018) | 47.07 | 66.28 | 97.68 | 46.67 | 55.25 | 87.33 |
| MISA (Hazarika et al., 2020) | 61.75 | 71.66 | 97.68 | 58.98 | 64.76 | 79.22 |
| UAVM (Gong et al., 2022) | 86.59 | 73.05 | 78.64 | 83.98 | **69.38** | 43.92 |
| DrFuse (Yao et al., 2024) | 66.83 | 75.35 | 97.68 | 62.86 | 69.33 | 78.56 |
| Perceiver (Jaegle et al., 2021) | 56.81 | 78.84 | 97.68 | 51.36 | 58.20 | 93.45 |
| AVoiD-DF (Yang et al., 2023) | 70.30 | 55.80 | 83.70 | 72.40 | 57.20 | 89.20 |
| VFD (Cheng et al., 2023) | - | - | 81.50 | - | - | 86.10 |
| Emo-Foren (Mittal et al., 2020) | - | - | 78.10 | - | - | 79.80 |
| MDS (Chugh et al., 2020) | - | - | 82.80 | - | - | 86.50 |
| rG-Concat(unimodal lin.) | 98.44 | 95.35 | - | 99.68 | 67.40 | - |
| rG-Concat | **95.74** | **95.35** | 97.68 | **99.51** | 56.40 | 98.70 |
| sG-Concat | 92.97 | **95.35** | 97.68 | 98.51 | 58.76 | 91.88 |
| sG-Perceiver | 64.01 | 82.15 | **98.84** | 66.53 | 62.42 | **99.31** |

Table 6: Comparison on the Food-101 data set. Fea. Dim represents the fusion input dimension, and Cls. represents the classification input dimension.

| Method | Fea. | Cls. | ACC (%) |
|---|---|---|---|
| ResNet-152 (He et al., 2016) | 2048 | 2048 | 65.45 |
| BERT (Devlin et al., 2019) | 768 | 768 | 86.71 |
| Concat | 2048-768 | 2816 | 78.33 |
| CMMP (Radford et al., 2021) | 768-768 | 768 | 83.45 |
| QMF (Zhang et al., 2023a) | 2048-768 | 1-1 | 79.10 |
| MLA (Zhang et al., 2024) | 768-768 | 768 | 85.14 |
| DrFuse (Yao et al., 2024) | 768-768 | 768 | 91.25 |
| MDF-FND (Lv et al., 2025) | 768-768 | 768 | 90.36 |
| sG-Concat | 2048-768 | 2816 | 91.52 |
| rG-Concat | 2048-768 | 2816 | 91.06 |
| sG-CMMP | 2048-768 | 768 | 93.23 |
| rG-CMMP | 2048-768 | 768 | 93.40 |
| rG-CMMP(image, lin.) | 768 | 768 | 67.21 |
| rG-CMMP(text, lin.) | 768 | 768 | 88.62 |

directly for linear probing, the "A" and "V" columns of the remaining methods represent another modal input of "0".

*Results.* As shown in Tab. 5, methods relying on contrastive objectives or disentanglement degrade under missing-modality and class-imbalance conditions, despite strong unimodal components. In contrast, PMR (rG-Concat and sG-Concat) consistently outperforms the concat baseline and prior fusion methods, with the largest gains under missing-modality. rPMR, emphasizing visual evidence, yields higher AUC when the audio modality is missing, while Perceiver-based sG-Perceiver shows similar trends, demonstrating PMR's ability to stabilize gradients and reduce sensitivity to imbalance without relying on any single modality. Further analysis reveals that unimodal linear probes outperform fusion models under missing-modality, with rG-CMMP showing significant improvement over unimodal probes in the visual modality. UAVM, which trains both modalities simultaneously, shows better performance with missing-modality but worse results with both modalities, indicating overfitting to the less informative modality.

### 5.3.2 FOOD-101: ADAPTING TO DIVERGENT FEATURE REQUIREMENTS

*Setup.* We evaluate image classification on Food-101 (Wang et al., 2015), where each image is paired with a detailed (and occasionally noisy) description. Text features are extracted by a pretrained BERT (Devlin et al., 2019) (768-d), and image features by a pretrained ResNet-152 (He et al., 2016) with global pooling (2048-d). To assess robustness to unequal feature lengths/dimensions, we compare PMR against QMF (Zhang et al., 2023a), MLA (Zhang et al., 2024), DrFuse (Yao et al., 2024), and MDF-FND (Lv et al., 2025); MLA uses a shared classifier for different-dimensional inputs, and QMF is adapted by replacing (1,3) pooling with global pooling. As a baseline for projection-based alignment, CMMP, DrFuse, and MDF-FND map 2048-d image features into the 768-d text space and apply an auxiliary loss. "(image/text lin.)" refer to linear probes applied to the image and text modalities, respectively, using features extracted from the frozen rG-CMMP encoders.

*Results.* QMF performs well with high-dimensional inputs but degrades with lower-dimensional or noisy modalities. MLA's enforced alignment can discard fine visual detail, and MDF-FND is sensitive to noise-induced distribution shifts. In contrast, G-CMMP explicitly separates shared and modality-specific components, filtering irrelevant cues while preserving discriminative visual information, improving accuracy under noisy annotations. These findings suggest that different tasks impose divergent requirements on feature components, and a flexible fusion prior is beneficial when feature dimensions and noise characteristics are mismatched. Furthermore, linear probing of the rG-CMMP encoders trained with paired supervision outperforms end-to-end fine-tuning of single-modality models, indicating that cross-modal gradients provide valuable guidance for within-modality representation learning.

### 5.4 ABLATION STUDY

**Structural ablation.** Tab. 7 contrasts: *Default* (full PMR: dissolve–dissociate–concentrate with conservation), *FFN* (D+C without dissociation/constraint; parameter-matched $d \to nd \to d$), *Linear*

Table 7: Ablation study on the Food101 dataset. $\mathcal{L}_{con}$ denotes the contrastive loss, "FFN" refers to the no-PMR feature reconfiguration ($d \rightarrow nd \rightarrow d$) structure (parameters are kept consistent), and "Linear" represents the process of mapping ResNet-152 output to 768d.

|  | Default | FFN | FFN+Linear | Linear+FFN | $\mathcal{L}_{con}$+Linear | $\mathcal{L}_{con}$+Linear+FFN | Linear | Identity |
|---|---|---|---|---|---|---|---|---|
| **Concat** | 91.52/91.06 | 80.61 | 80.50 | 80.93 | 82.30 | 86.63 | 78.29 | 78.33 |
| **CMMP** | 93.23/93.40 | - | 81.33 | 81.01 | 83.45 | 87.15 | 79.35 | - |

(C-only), and *Identity* (no reconfiguration), plus ordering variants (*FFN+Linear/Linear+FFN*) and contrastive replacements ($\mathcal{L}_{con}+\cdot$). Across settings, removing *dissociation+conservation* consistently degrades outcomes, while parameter-matched reconfiguration or contrastive auxiliaries do not recover the gap. This stagewise proxy covers C-only, D+C (no dissociation), and no-reconfiguration, indicating that PMR's gains stem from the single-potential with conservation—i.e., enforcing separation as reordering rather than discard—rather than from extra layers or parameter count.

**Parameter sensitivity.** Fig. 3 varies the effective capacity of PMR via the first component $n \times b$ (expansion $n$ times the learned dissociation boundary ratio $b$). When $n \times b < 1$, the dissociation channel is under-parameterized and accuracy is sensitive; once $n \times b \geq 1$, the curve flattens, indicating sufficient degrees of freedom for separation and hence weak sensitivity. In contrast, Concat is highly sensitive to the chosen dimensionality of the specific branch, reflecting its tendency to amplify modality-specific patterns; PMR, under a single-potential with conservation, is far less affected by this knob.

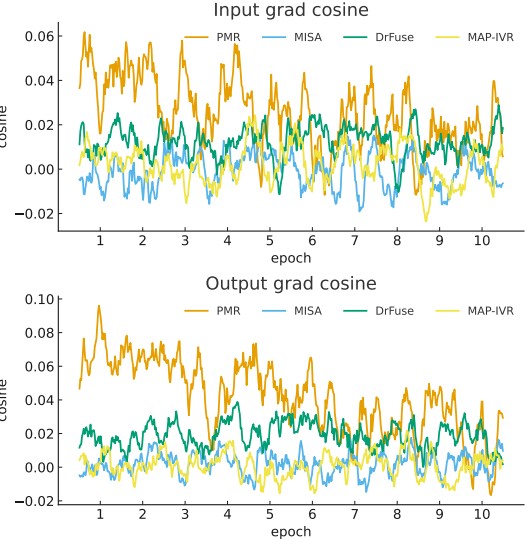

Figure 3: Hyper-parameter sensitivity to $b$ and $n$ on the Food101 dataset.

## 5.5 VISUALIZATION

This subsection visualizes the inner products between the gradients of the main and auxiliary losses at the encoder input and output. As shown in Fig. 4, PMR differs from disentanglement baselines by optimizing a unified objective that couples similarity with reconstruction, under an information-preservation constraint. This reduces cross-loss gradient conflict, whereas MISA (Hazarika et al., 2020), DrFuse (Yao et al., 2024), and MAP-IVR (Liu et al., 2021a) use orthogonality, exclusivity, and Cross-Entropy terms that often conflict under label and alignment noise. At the encoder input, gradients are noise-dominated and near-isotropic, while PMR achieves agreement more steadily. At the output, PMR leads to a more stable, aligned geometry, preserving robustness without overpowering the downstream head. Initially, PMR aligns with the downstream loss, and by convergence, the mean approaches 0, indicating effective reduction of information entropy and timely optimization dominance.

Figure 4: Gradient cosine on VGGSound.

## 6 CONCLUSION

PMR unifies task and information-preservation into a single scalar-potential objective acting on one shared reconfiguration block (dissolve–dissociate–concentrate), enforcing conservation so separation is reordering rather than discard, thereby delivering robust, interpretable multimodal fusion. Limitation and future work see Appendix B.

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

# A    EXPERIMENTS DETAILS

**Hardware.** RTX 4090 GPU @ 2.64GHz with 24GB V-RAM, running on AMD Ryzen-9 5900X @ 4.70GHz CPU with 64GB System RAM. Operating System is Ubuntu 20.04.

**PMR variant.** PMR-MoE denotes attaching an additional mixture-of-experts (MoE) module at the shared output of PMR. The MoE architecture follows Yuan et al. (2025), with a hidden-layer dimension of $2l^*$, and comprises 1 of shared experts and 4 groups of experts. PMR-SRHT denotes using the Subsampled Randomized Hadamard Transform (SRHT) for dimensionality expansion and reduction; rPMR computes the reconstruction error in the input domain, whereas sPMR computes similarity/dissimilarity in the SRHT domain.

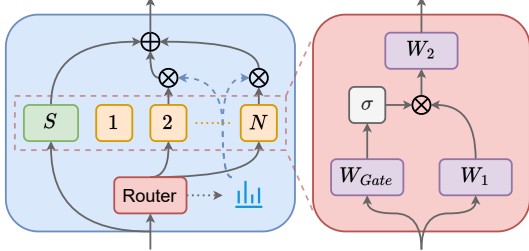

Figure 5: MoE structure.

**Reproductable.** For all experiments, we use apex to optimize the v-memory and the parameter is set to 'O1'. For VGGSound, using bf16. The random seed fixed '1' for all PMR related implementation. For VGGSound, random seed fixed '42' for pretrain.

(1) torch.manual_seed(seed)

(2) torch.cuda.manual_seed_all(seed)

(3) np.random.seed(seed)

(4) random.seed(seed)

## A.1    MODALITY EXTRACTOR

**Text for Food-101.**

1. Tokenizer: AutoTokenizer(textattack/bert-base-uncased-yelp-polarity).

2. BERT: BertForSequenceClassification(textattack/bert-base-uncased-yelp-polarity).

**Image.**

1. ActivityNet: Feature from APIVR (Xu et al., 2020).

2. Food-101: ResNet-152 (He et al., 2016) pretrained from torch (Paszke et al., 2019). Resize to 256×256, center crop 224×224, normalization, without any augmentation.

**Audio.**

1. FakeAVCeleb: 0.8s sample length, R(2+1)D (Tran et al., 2018) as extractor, preprocess via log(abs(STFT(fft=1024, hop=256, win length=1024,window=blackman(1024)))+1e-8)

2. FakeAVCeleb: 9.0s sample length, ResNet34 (He et al., 2016) as extractor, preprocess via log(abs(STFT(fft=1024, hop=256, win length=1024,window=blackman(1024)))+1e-8)

**Video.**

1. FakeAVCeleb: 8 frames, R(2+1)D (Tran et al., 2018) as extractor. Resize to 224, center crop, ToTensor() normalization.

2. VGGSound: 30 frames (Gong et al., 2022), ResNet34 (He et al., 2016) as extractor, temporal average.

3. ActivityNet: Feature from APIVR (Xu et al., 2020).

Table 8: Details of PMR. Momentum of SGD = 0.9, weight decay = $1e-4$. AdamW betas = $(0.9, 0.999)$, weight decay = $5e-2$.

| Dataset | Lr | Optimizer | Batchsize | Epoch | Input Shape |
|---|---|---|---|---|---|
| ActivityNet | $1e^{-2}$ | SGD | 64 | 20 | [4096,128], [128,128] |
| VGGSound | $1e^{-4}$ | AdamW | 256 | 10 | [512,512] |
| FakeAVCeleb | $1e^{-2}$ | SGD | 64 | 20 | [512,512] |
| Food-101 | $1e^{-4}$ | AdamW | 16 | 10 | [2048,768] |

## A.2 IMPLEMENT DETAILS

More details are listed in Table 8.

Method based on SGD use MultiStep LR Scheduler, based on Adawm use Cosine LR Scheduler. Furthermore, PMR use a additional SGD optimizer, with $lr = 2.5e^{-4}$, momentum = 0.9, weight decay = $1e-4$. In our default setting, $b = 0.5$, $n = 4$. This is almost consistent with the FFN structure. In practice, the boundary $b(m)$ should be understood as an effective interface that occupies a finite interval rather than a sharply defined point. This is analogous to the separation layer produced in continuous PNP dynamics under strong external driving, where species distributions form a pronounced yet diffuse transition zone. In the discrete feature space of neural networks, tensor dimensions are inherently quantized, so selecting a representative boundary $b$ and ensuring that each dissolved segment maintains sufficiently large dimensionality (no smaller than its original allocation) is already sufficient to preserve conditional full rank and thereby guarantee reconstructability.

## B LIMITATION AND FUTURE WORK

Standard PMR is implemented with an FFN style rearrangement block of complexity $O(d^2)$. Although a simple SRHT based variant already attains competitive performance, exploring low rank parameterizations remains a promising direction. The current PMR is time independent, and our evaluation follows prior benchmarks for fair comparison; therefore we omit settings that emphasize temporal modeling or newer downstream objectives. Future work will introduce temporal structure in order to leverage the time dependent Poisson-Nernst-Planck formulation and will assess PMR on sequence modeling tasks. Finally, because the PMR objective is task conditioned, the same scalar potential formulation can instantiate self supervised pretext losses as $L_{\text{task}}$; developing such self supervised variants is a planned direction.

## C INFERENCE AND REDEFINITION OF MULTIMODAL TASK OBJECTIVES

In this section, we present an entropy-based auxiliary objective that biases task-loss gradients toward the earliest extractors, encouraging them to produce maximally task-informative features. Specially, supervised task loss aligns model predictions with ground-truth labels. In unimodal settings, its gradient flows directly to the feature extractor, but multimodal architectures introduce a fusion stage whose loss can conflict with the primary task objective. Conventional approaches rarely manage this gradient interaction, allowing fusion objectives to perturb extractor updates. As shown in Fig. 6, multimodal tasks involve three key parameter sets: feature extractors $\theta = \{\theta^{(1)}, \theta^{(2)}, \ldots, \theta^{(M)}\}$, fusion parameters $\theta^F$, and task head parameters $\theta^C$. The fusion module $g(\cdot)$ extends the output of the feature extractors $f(X, \theta)$, leading to the learning objective

$$\min_{\theta, \theta^F} H(g(f(X, \theta), \theta^F) \mid f(X, \theta)) \tag{7}$$

Similarly, with $h(\cdot)$ mapping features to the downstream task, the objective becomes

$$\min_{\theta, \theta^F, \theta^C} H(Y \mid h(g[f(X, \theta), \theta^F], \theta^C)).$$

Because the gradients from the feature extractors pass through $g(\cdot)$, the optimization is split into:

$$\min_{\theta^F, \theta^C} H(Y \mid h(g[f(X, \theta), \theta^F], \theta^C)) \tag{8}$$

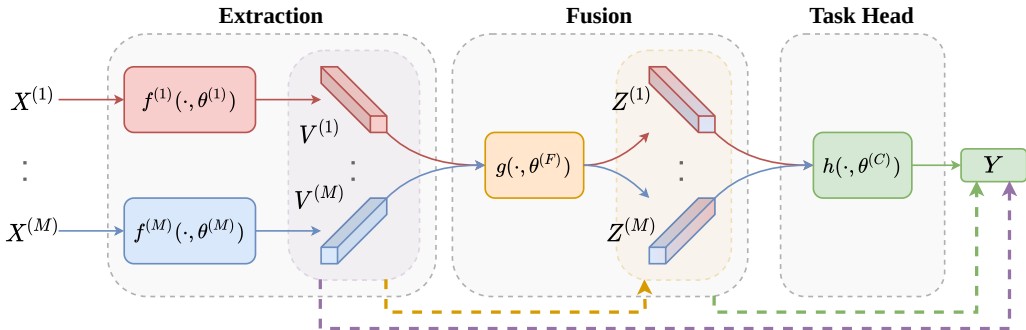

Figure 6: Stages of information entropy change. The output features are set to correspond with the input features, ensuring a unique task outcome. The light yellow and light green arrows correspond to Eq. (7) and Eq. (8), respectively, while the light purple arrow corresponds to Eq. (9).

and Eq. (7). In this way, any loss from the fusion stage properly adjusts its parameters. Feature extraction through dimensionality reduction reduces data uncertainty measured by $H(\cdot)$. For a single modality, the feature extractor $f^{(m)}(\cdot, \theta^{(m)})$ and task head $h^{(m)}(\cdot, \theta^{h^{(m)}})$ are directly optimized via

$$\min_{\theta^{(m)}, \theta^{h^{(m)}}} H\Big[Y \mid h^{(m)}(f^{(m)}(X^{(m)}, \theta^{(m)}), \theta^{h^{(m)}})\Big],$$

compressing data samples into a feature space that retains task-relevant attributes. Viewing loss as a tool for entropy reduction helps maximize the mutual information among related features. For multimodal fusion, the aim is to minimize the entropy of fused features relative to the sum of the entropies of individual modalities. The joint entropy of the system can be written as

$$H(f(X,\theta)) = \sum_{m=1}^{M} H\big(f^{(m)}(X^{(m)}, \theta^{(m)})\big) - I(f(X,\theta)), \quad \min_{\theta} H(f(X,\theta)) \Leftrightarrow \max_{\theta} I(f(X,\theta)),$$

where $I(f(X,\theta))$ is mutual information. Downstream objectives typically minimize mutual information, thereby reducing entropy. However, the difference between Eqs. (7) and (8) suggests that some fusion methods may not maintain a direct correspondence between inputs and outputs. Achieving complete consistency (i.e., zero mutual information) does not always yield the best performance and can even increase the entropy of downstream features (Wang & Liu, 2020; Liang et al., 2022; Jiang et al., 2023; Du et al., 2023)—as seen in methods like AVoiD-DF (Yang et al., 2023) and MBT (Nagrani et al., 2021), which suffer from a reduced ability to extract modality-specific features when modalities are missing. Therefore, the optimization objectives for multimodal tasks must balance minimizing entropy during fusion while ensuring that the entropy of downstream features is not increased, with the overall goal of minimizing entropy in the fusion stage while preserving or reducing the entropy of features relevant to the downstream task:

$$\min_{\theta, \theta^F, \theta^C} \{H(Y \mid h(g[f(X,\theta), \theta^F], \theta^C)])\}$$

$$\text{s.t.} \quad \forall m \in \{1, 2, \ldots, M\}, \quad \theta^{(m)} \in \arg\min_{\hat{\theta}} H\big(Y \mid f^{(m)}(X^{(m)}, \hat{\theta})\big) \tag{9}$$

## D  SCALAR POTENTIAL: PHYSICAL (VARIATIONAL) AND NEURAL DEFINITIONS

This subsection defines scalar potential and explains the differences between PMR and the decoupling of mutual information loss operations.

**Definition 1** (Scalar potential in continuum physics / gradient-flow systems (Ambrosio et al., 2008; Jordan et al., 1998; Jüngel, 2016; Markowich et al., 1990)). Let $\mathcal{X}$ be a state space (e.g., nonnegative concentrations $c = (c_i)_{i=1}^{K}$) and let $\mathcal{F} : \mathcal{X} \to \mathbb{R}$ be a differentiable free-energy functional. A *scalar potential* is such $\mathcal{F}$ for which the evolution is a (generalized) gradient flow

$$\partial_t c = \nabla \cdot \Big(M(c) \nabla \frac{\delta \mathcal{F}}{\delta c}\Big),$$

with mobility $M(c) \succeq 0$, yielding the energy–dissipation and conservation laws

$$\frac{\mathrm{d}}{\mathrm{d}t}\,\mathcal{F}(c(t)) \;\leq\; 0, \qquad \frac{\mathrm{d}}{\mathrm{d}t}\int_\Omega c_i\,\mathrm{d}x \;=\; 0 \quad (i=1,\dots,K).$$

For Poisson–Nernst–Planck/Drift–Diffusion models, a standard choice is

$$\mathcal{F}(c,\phi) \;=\; \sum_{i=1}^{K}\int_\Omega k_B T\, c_i(\log c_i - 1)\,\mathrm{d}x \;+\; \frac{\varepsilon}{2}\int_\Omega |\nabla\phi|^2\,\mathrm{d}x \;+\; \sum_{i=1}^{K} z_i e \int_\Omega c_i\,\phi\,\mathrm{d}x,$$

together with $-\nabla\cdot(\varepsilon\nabla\phi) = \sum_i z_i e\, c_i$ and fluxes $J_i = -D_i\big(\nabla c_i + \frac{z_i e}{k_B T} c_i \nabla\phi\big)$, which realizes the gradient-flow/Onsager structure and the dissipation inequality (Jüngel, 2016; Markowich et al., 1990).

**Definition 2** (Scalar potential in neural optimization / blockwise scalarization (Miettinen, 1999; Marler & Arora, 2004; LeCun et al., 2006; Sener & Koltun, 2018)). Let $\theta = (\theta_E, \theta_R, \theta_H)$ be network parameters, and let $\{L_i(\theta)\}_{i=1}^m$ be differentiable objectives. A *neural scalar potential* is a weighted sum

$$\Phi(\theta) \;=\; \sum_{i=1}^{m} \lambda_i\, L_i(\theta), \quad \lambda_i \geq 0,$$

whose steepest-descent field $-\nabla_\theta \Phi$ defines the unique update direction. On a shared bottleneck $Z = f_R(V; \theta_R)$, the blockwise condition

$$\nabla_{\theta_R}\Phi \;=\; J_R^\top \sum_{i=1}^{m} \lambda_i\, \frac{\partial L_i}{\partial Z}, \qquad J_R := \frac{\partial Z}{\partial \theta_R},$$

ensures a *single* descent direction on $\theta_R$ in each step. Under joint convexity, minimizers of $\Phi$ with $\lambda \in \Delta^{m-1}$ are (weakly) Pareto-optimal and every Pareto point solves some $\min_\theta \sum_i \lambda_i L_i(\theta)$ (weighted-sum scalarization) (Miettinen, 1999; Marler & Arora, 2004). Energy-based learning is a special case with a single scalar energy $E_\theta(x)$ driving inference and learning (LeCun et al., 2006).

Because in PMR both losses traverse the same bottleneck $Z = f_R(V; \theta_R)$ and thus update the same rearrangement parameters via the single step

$$-\nabla_{\theta_R}\Phi = -J_R^\top\Big(\frac{\partial L_{\text{task}}}{\partial Z} + \lambda\,\frac{\partial L_{\text{pres}}}{\partial Z}\Big), \quad J_R := \frac{\partial Z}{\partial \theta_R},$$

so a scalar potential holds on $\theta_R$; by contrast, in generic decoupling the auxiliary terms attach to encoder-side modules while the task loss attaches to heads, yielding $\frac{\partial L_{\text{aux}}}{\partial \theta_R} \approx 0$ on the fusion block and hence no single-block scalar potential in one step (Sener & Koltun, 2018).

# E  POISSON–NERNST–PLANCK PERSPECTIVE ON FEATURE DISSOCIATION

## E.1  INTUITION

The Poisson–Nernst–Planck (PNP) equations describe how positively and negatively charged particles move inside an electrolyte until the system reaches electro–diffusive *equilibrium*. In our analogy, multimodal *features* play the role of those charged particles. e.g.:

- Each *shared* feature corresponds to a "cation",
- Each *modality-specific* feature corresponds to an "anion".

An external voltage $\phi_{\text{ext}}$ (implemented as a learnable loss term) pulls the two species towards opposite ends of a one–dimensional 'cell' that represents the feature axis. Because the PNP system *conserves charge* (mass conservation law), it can reorganise the mixture *without* changing the total amount of information. Hence, after training, features self-separate into two nearly disjoint blocks—exactly what is required for flexible multimodal fusion.

## E.2 EXISTENCE OF AN EQUILIBRIUM INTERFACE

Let the cell length be $l$ (with $l > \lambda_D$, the Debye length) and denote concentrations by $c_\oplus(x)$ and $c_\ominus(x)$. Under the steady-state PNP equations

$$\mathbf{J}_i = -D_i \nabla c_i + \tfrac{D_i z_i e}{k_B T} c_i \nabla \phi, \quad \nabla^2 \phi = -\tfrac{e}{\varepsilon} \sum_i z_i c_i,$$

there always exists a point $b \in [0, l]$ such that

$$c_\oplus(b) = c_\ominus(b).$$

Around $b$ the net charge is zero, so the electric field changes sign, forming a natural "interface" that cleanly divides the two species. When $l \ll \lambda_D$ or when the applied voltage $U_0 = \phi_{\text{ext}}(0) - \phi_{\text{ext}}(l)$ is sufficiently large, the steady-state profiles approach step functions: positive ions cluster near $x = 0$, negative ions near $x = l$. These statements can be proved by integrating Gauss's law from 0 to $b$:

$$\Big[\partial_x \phi\Big]_{x=0}^{x=b} \approx -\frac{e}{\varepsilon_0} \int_0^b z_\ominus c_\ominus(x)\, dx.$$

Our numerical solver attains a residual below $1 \times 10^{-8}\,\%$.

## E.3 WHY PNP WORKS—FOUR KEY CLAIMS EXPLAINED

Below we restate the four open issues from Sec. 1 and clarify, with the PNP analogy, *why* they are resolved.

**1) *Fixed feature dimensions cause redundancy* (Ying, 2019; Yao et al., 2024; Liu et al., 2021b).** Because the PNP flux automatically moves charges until the local electric field vanishes, the final *amount* of each species is determined by electro-diffusive balance, *not* by the initial container size. Translating back, our loss encourages the network to *re-allocate* channel capacity: redundant shared components "flow" away from modality-specific components, reducing overlap without enforcing equal dimensionality. Specially

$$w_{\text{specific}}^{(m)} e = -\frac{d_{\text{shared}}}{d_{\text{specific}}^{(m)}} w_{\text{shared}} e, \tag{10}$$

where $e$ is the primary charge, $d_{\text{shared}}$ is the shared feature length, $d_{\text{specific}}^{(m)}$ is the specific feature length of modality $(m)$, and $w_{\text{shared}}, w_{\text{specific}}^{(m)}$ are the corresponding calculable equivalent charge quantities (weights) respectively.

**2) *Assuming complete inputs degrades performance when modalities are missing* (Ma et al., 2022; Wang et al., 2023a; Yao et al., 2024).** Mass conservation in PNP guarantees a well-defined steady state even if the initial concentration of one species is zero. Likewise, when an entire modality is absent (its concentration is momentarily zero), the remaining features re-equilibrate through the same loss, so the model degrades gracefully instead of collapsing.

**3) *Uniform features cannot serve diverse downstream tasks* (Xu et al., 2020; Liu et al., 2021a; Xia et al., 2023).** In PNP the external potential $\phi_{\text{ext}}$ is a boundary condition. Changing $\Delta\phi_{\text{ext}}$ instantly reshapes the equilibrium profiles *without* redefining particle types (Eq.(6)). Our framework mirrors this: adjusting the task loss selectively tilts the "voltage" so that task-relevant features migrate into the most informative sub-space, capturing modality-specific details while still sharing global context.

**4) *Auxiliary losses may introduce conflicting gradients* (Yang et al., 2023; Hazarika et al., 2020; Wang & Liu, 2020).** All PNP forces derive from a *single scalar potential*; their gradients are therefore conservative and cannot oppose each other. By casting every auxiliary objective as a contribution to $\phi_{\text{ext}}$, we ensure that the resulting gradients are *additive* in potential space and hence orthogonalised in feature (PNP's) space, mitigating destructive interference.

These four properties emerge directly from the physics of charge conservation and electro-diffusion and therefore hold independently of the specific network architecture.

### E.4 Supplement of Methodology

To apply our analogy to neural networks, we designed a set of loss functions that simulate changes in external potential. Specifically, we employed the dissociation matrix ($\mathbf{P}_{\text{dis}}^{(m)}$) and concentration matrix ($\mathbf{P}_{\text{shared}}^{(m)}$, $\mathbf{P}_{\text{specific}}^{(m)}$) to guide the separation and reconstruction of features. For each modality, the dissociation matrix maps the feature representation into a higher-dimensional space, analogous to dissolution in physics. The concentration matrix applied feature dimensionality reduction to eliminate irrelevant information, such as "solvent," analogous to the condensation process in physical systems. Therefore, loss function $\mathcal{L}_{\text{preserve}}$ ensures that information is preserved during this mapping, similar to mass conservation in physical systems. The combined use of these matrices allows features to migrate and separate while maintaining overall information integrity. The mass conservation loss function mandates that features migrate as completely as possible, regardless of their current state. By simulating the changes in potential through this loss function, features that have not fully migrated are driven towards the target region, ultimately achieving effective feature reconstruction and separation.

To formalize this, consider the mapping,

$$\hat{Z}_i^{(m)} = \mathbf{P}_{\text{dis}}^{(m)} V_i^{(m)}, \quad \hat{Z}_{i,\text{shared}}^{(m)} = \hat{Z}_i^{(m)}(1:b^{(m)}), \quad \hat{Z}_{i,\text{specific}}^{(m)} = \hat{Z}_i^{(m)}(b^{(m)}+1:n\,l^{(m)}).$$

Subsequently, information conservation is calculated based on the concentrated results

$$\mathcal{L}_{\text{preserve}} = \text{Sim}\Big[\Big(\big[\mathbf{P}_{\text{shared}}^{(m)} \hat{Z}_{i,\text{shared}}^{(k)}\big]; \big[\mathbf{P}_{\text{specific}}^{(m)} \hat{Z}_{i,\text{specific}}^{(m)}\big]\Big), V_i^{(m)}\Big], \tag{11}$$

where $\text{Sim}(\cdot)$ represents a similarity calculation function (for example, Euclidean distance or cosine similarity). Here, $l^{(m)}$ and $b^{(m)}$ denote the feature dimension and dissociation boundary of modality $m$, respectively. Near this boundary, features are distinctly separated. The mapping matrices $\mathbf{P}_{\text{dis}}^{(m)} \in \mathbb{R}^{nl^{(m)} \times l^{(m)}}$, $\mathbf{P}_{\text{shared}}^{(m)} \in \mathbb{R}^{l^* \times b^{(k)}}$, $\mathbf{P}_{\text{specific}}^{(m)} \in \mathbb{R}^{l^{(m)} \times (nl^{(m)} - b^{(m)})}$, and $\mathbf{P}_{\text{recon}}^{(m)} \in \mathbb{R}^{l^{(m)} \times (l^{(m)} + l^*)}$ are learnable parameters. $\hat{Z}_i^{(m)} \in \mathbb{R}^{nl^{(m)}}$ is the result of linearly mapping (dissociating) $V_i^{(m)} = f^{(m)}(X_i^{(m)}, \theta^{(m)}) \in \mathbb{R}^{l^{(m)}}$ to a higher-dimensional space.

Due to the balance between diffusion and electromigration, the system ultimately reaches a steady state. The external loss function acts as an external electric field, driving feature particles (ions) to move and aggregate in different directions. According to the law of mass conservation, the original modality representations can be completely reconstructed at the other end. The reconstruction loss effectively simulates particle movement, and its reduction does not lead to ambiguity due to the presence of modality-specific features.

The parameter count and computational load of the standard PMR (P implemented using Linear) are shown in Tab. 9. The SRHT variant introduces no learnable parameters; For the MoE (shared-only) variant, the additional parameter count is

$$\Delta\text{Params}_{\text{MoE}} = \Big[(\tfrac{8}{3}l^* \times l^*) + 2 * (\tfrac{8}{3}l^* \times l^*)\Big] \times (4+1) = 20\,(l^*)^2,$$

where 4 denotes the gated experts, 1 the shared expert, and $\tfrac{8}{3}l^*$ the hidden width. The corresponding increase in computation is

$$\Delta\text{FLOPs} \approx 16\,(l^*)^2 + 10\,l^*.$$

## F Loss Function from the External Excitation Perspective

The second law of thermodynamics indicates that the total entropy of a system tends to increase, indicating that the system naturally tends toward disorder (Zemansky & Menger, 1952). Thermodynamic

Table 9: Parameter count and computational complexity for PMR (Linear Mapping) matrices per modality

| Modality $m$ | | Parameters | FLOPs |
|---|---|---|---|
| **Matrix** | $\mathbf{P}_{\text{dis}}^{(m)}$ | $n \cdot l^{(m)} \times l^{(m)} + n \cdot l^{(m)}$ | $2n \cdot (l^{(m)})^2 + n \cdot l^{(m)}$ |
| | $\mathbf{P}_{\text{shared}}^{(m)}$ | $b^{(m)} \times l^* + b^{(m)}$ | $2b^{(m)}l^* + b^{(m)}$ |
| | $\mathbf{P}_{\text{specific}}^{(m)}$ | $(nl^{(m)} - b^{(m)}) \times l^{(m)} + (nl^{(m)} - b^{(m)})$ | $2(nl^{(m)} - b^{(m)})l^{(m)} + (nl^{(m)} - b^{(m)})$ |
| | $\mathbf{P}_{\text{recon}}^{(m)}$ | $l^{(m)} \times (l^{(m)} + l^*) + l^{(m)}$ | $2l^{(m)}(l^{(m)} + l^*) + l^{(m)}$ |

entropy, in statistical mechanics, is expressed as

$$S = -k_B \sum_i p_i \ln p_i,$$

with $k_B$ being Boltzmann's constant and $p_i$ the probability of the system being in microstate $i$. However, during the training process in machine learning, the loss function can be likened to external entropy-reducing excitation, driving the system toward a more ordered and optimized state (Shang et al., 2021). For example, the formula for cross-entropy loss is

$$\mathcal{L}_{\mathrm{CE}} = -\frac{1}{N} \sum_{i=1}^{N} \sum_{k=1}^{K} y_{i,k} \log(\hat{y}_{i,k}),$$

where $N$ is the number of samples, $K$ is the number of classes, $y_{i,k}$ is the true label indicating whether sample $i$ belongs to class $k$ (usually in one-hot encoding), and $\hat{y}_{i,k}$ is the predicted probability that sample $i$ belongs to class $k$. Cross-entropy loss directly reduces the model's information entropy by minimizing the difference between the predicted distribution and the true distribution. This reduction of information entropy is analogous to the reduction of entropy increase in a thermodynamic system under external influence, leading the system to a more ordered state, Information entropy is defined as

$$H(X) = -\sum_x p(x) \log p(x),$$

where $p(x)$ is the probability of event $x$. The mathematical similarity between $H(X)$ and $S$ indicates that both quantify uncertainty in probability distributions, albeit in different contexts. In physics, external excitations like electric potentials drive particles to move in specific directions, forming ordered structures. Similarly, loss functions in machine learning guide model parameters through optimization algorithms to reduce prediction uncertainty and improve accuracy. Specifically, cross-entropy loss acts as an external excitation, transitioning the model from a high-entropy (disordered) state to a low-entropy (ordered) state. This process is akin to guiding ions in an electric field to reduce system disorder, as illustrated in Fig. 7.

Viewing the loss function as an external excitation provides insight into how optimization drives a system toward order, effectively managing uncertainty to enhance model performance. This perspective informs network design, where the loss function serves as a driving force that guides feature separation in a manner analogous to particle movement under external forces, shift inspires the design of physics-inspired models.

## G  DERIVATION OF THE APPROXIMATE LAW

For particles $i \in \{\oplus, \ominus\}$, the total flux $\mathbf{J}_i$ comprises three components: diffusion, convection, and electromigration. The standard vector form of the Nernst-Planck equation is:

$$\mathbf{J}_i = -D_i \nabla c_i + c_i \mathbf{v} + \frac{D_i z_i e}{k_B T} c_i \mathbf{E} \tag{12}$$

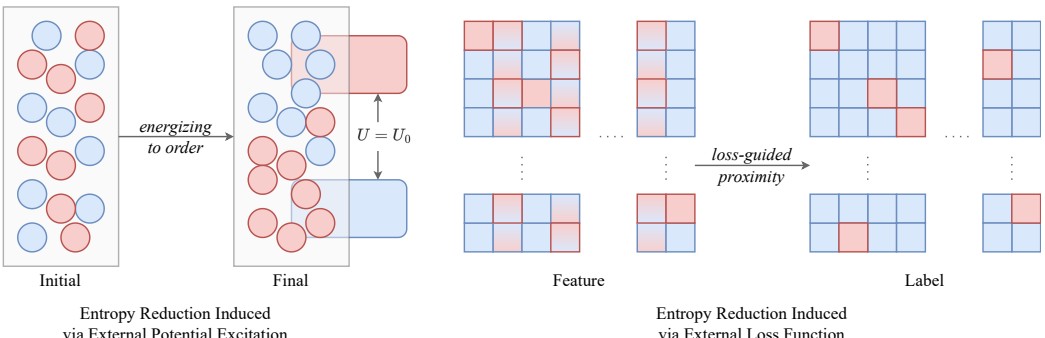

Figure 7: Entropy reduction in different systems. Left: ions align under an external electric field (red and blue represent opposite charges); Right: cross-entropy loss guides classifier outputs toward labels (red approaching 1, blue approaching 0).

where $\mathbf{J}_i$ is the flux of particle $i$, $D_i$ is the diffusion coefficient of particle $i$, $c_i(\mathbf{r}, t)$ is the concentration of particle $i$ at position $\mathbf{r}$ and time $t$, $\mathbf{v}$ is the fluid velocity, $z_i$ is the charge number of particle $i$, $e$ is the elementary charge, $k_B$ is the Boltzmann constant, $T$ is the absolute temperature, and $\mathbf{E}$ is the electric field intensity.

According to Maxwell's equations, the electric field $\mathbf{E}$ is given by

$$\mathbf{E} = -\nabla\phi - \frac{\partial \mathbf{A}}{\partial t},$$

where $\phi$ is the electric potential and $\mathbf{A}$ is the magnetic vector potential. In the proposed model, it is assumed that there is no time-varying magnetic field ($\frac{\partial \mathbf{A}}{\partial t} = \mathbf{0}$), simplifying the electric field to

$$\mathbf{E} = -\nabla\phi.$$

This indicates that the electric field $\mathbf{E}$ is generated by the gradient of the electric potential $\phi$.

In systems with external potentials applied, the total electric potential $\phi$ is considered as the sum of the internal potential $\phi_{\text{int}}$, due to the charge distribution within the system, and the external potential $\phi_{\text{ext}}$, imposed by external sources:

$$\phi = \phi_{\text{int}} + \phi_{\text{ext}}.$$

The external potential $\phi_{\text{ext}}$ establishes boundary conditions and influences the distribution of the internal potential $\phi_{\text{int}}$. The electric field becomes

$$\mathbf{E} = -\nabla\phi = -\nabla(\phi_{\text{int}} + \phi_{\text{ext}}) = -\nabla\phi_{\text{int}} - \nabla\phi_{\text{ext}}. \tag{13}$$

Starting from the mass conservation continuity equation,

$$\frac{\partial c_i}{\partial t} = -\nabla \cdot \mathbf{J}_i,$$

where $c_i$ is the concentration of ion species $i$ and $\mathbf{J}_i$ is the flux of ion $i$. Under steady-state conditions ($\frac{\partial c_i}{\partial t} = 0$), this simplifies to

$$\nabla \cdot \mathbf{J}_i = 0.$$

Given the assumptions of zero fluid velocity ($\mathbf{v} = \mathbf{0}$) and no time-varying magnetic field, the particle flux simplifies to the Nernst-Planck equation:

$$\mathbf{J}_i = -D_i \nabla c_i - \frac{D_i z_i e}{k_B T} c_i \nabla\phi,$$

where $D_i$ is the diffusion coefficient, $z_i$ is the charge number of ion $i$, $e$ is the elementary charge, $k_B$ is Boltzmann's constant, and $T$ is the absolute temperature. The total potential $\phi$ is used here because the movement of ions depends on the total electric field. In the one-dimensional case along the $x$ direction, the flux becomes

$$J_i = -D_i \left( \frac{dc_i}{dx} + \frac{z_i e}{k_B T} c_i \frac{d\phi}{dx} \right). \tag{14}$$

Since $\nabla \cdot \mathbf{J}_i = 0$, implying $\frac{dJ_i}{dx} = 0$, and assuming no net ionic flux ($J_i = 0$), Eq. (14) becomes

$$0 = -D_i \left( \frac{dc_i}{dx} + \frac{z_i e}{k_B T} c_i \frac{d\phi}{dx} \right). \tag{15}$$

Rewriting Eq. (15), we obtain

$$\frac{dc_i}{dx} = -\frac{z_i e}{k_B T} c_i \frac{d\phi}{dx}. \tag{16}$$

This equation describes how the ion concentration $c_i(x)$ varies with position, related to the gradient of the total electric potential $\frac{d\phi}{dx}$.

Integrating Eq. (16) yields

$$\ln\left( \frac{c_i(x)}{c_i^0} \right) = -\frac{z_i e}{k_B T} \left( \phi(x) - \phi^0 \right), \tag{17}$$

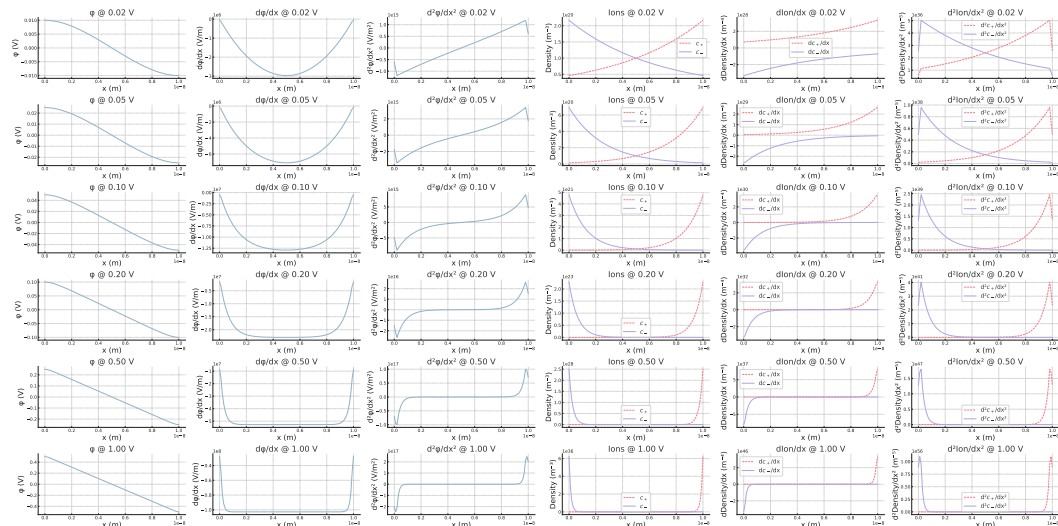

Figure 8: Newton-Raphson iteration results. Defined constants: $l = 1 \times 10^{-8}$ m (electrolyte length), $c_0 = 1 \times 10^{20}$ m$^{-3}$ (ion concentration), $T = 300$ K (absolute temperature), $e_{\text{charge}} = 1.602 \times 10^{-19}$ C (elementary charge), $k_B = 1.381 \times 10^{-23}$ J/K (Boltzmann constant), and $\epsilon_0 = 8.854 \times 10^{-12}$ F/m (vacuum permittivity).

where $c_i^0 = c_i(x_0)$ and $\phi^0 = \phi(x_0)$ are the concentration and potential at a reference position $x_0$. Choosing $\phi^0 = 0$ and $x_0 = 0$, we obtain

$$c_i(x) = c_i^0 \exp\left(-\frac{z_i e}{k_B T}\phi(x)\right).$$

The internal electric potential $\phi_{\text{int}}(x)$ satisfies Poisson's equation:

$$\frac{d^2\phi_{\text{int}}(x)}{dx^2} = -\frac{\rho(x)}{\varepsilon_0}, \tag{18}$$

where $\varepsilon_0$ is the vacuum permittivity, and $\rho(x)$ is the spatial charge density given by

$$\rho(x) = e\sum_i z_i c_i(x).$$

Here, $c_i(x)$ depends on the total potential $\phi(x)$ as per Eq. (17). Substituting $c_i(x)$ into Poisson's equation Eq. (18), we have

$$\frac{d^2\phi_{\text{int}}(x)}{dx^2} = -\frac{e}{\varepsilon_0}\sum_i z_i c_i^0 \exp\left(-\frac{z_i e}{k_B T}\phi(x)\right). \tag{19}$$

Introducing the dimensionless potential $u(x) = \frac{e\phi(x)}{k_B T}$, we can rewrite the equation as

$$\frac{d^2 u_{\text{int}}(x)}{dx^2} = -\frac{1}{\lambda_D^2}\sum_i \frac{z_i c_i^0}{\sum_j z_j^2 c_j^0} z_i \exp\left(-z_i u(x)\right), \tag{20}$$

where the Debye length $\lambda_D$ is defined as

$$\lambda_D = \sqrt{\frac{\varepsilon_0 k_B T}{e^2 \sum_i z_i^2 c_i^0}}.$$

For general ion valences $z_\oplus$ and $z_\ominus$, this equation remains complex and lacks an analytical solution; therefore, numerical methods are employed. In numerical simulations of Eq. (20), the Newton-Raphson iterative method can be used, as shown in Figure 8.

To find the point $x = b$ where the concentrations of positive and negative ions are equal, define the function
$$f(x) = c_\oplus(x) - c_\ominus(x).$$

Using Eq. (17), $f(x)$ becomes
$$f(x) = c_\oplus^0 \exp\left(-\frac{z_\oplus e}{k_B T}\phi(x)\right) - c_\ominus^0 \exp\left(-\frac{z_\ominus e}{k_B T}\phi(x)\right).$$

Assuming equal initial concentrations ($c_\oplus^0 = c_\ominus^0$) and valences $|z_\oplus| = |z_\ominus|$, and considering the behavior of $\phi(x)$ due to the external potential, we have $f(0) < 0$ and $f(l) > 0$. By the Intermediate Value Theorem, there exists a point $b \in (0, l)$ such that $f(b) = 0$, implying
$$c_\oplus(b) = c_\ominus(b). \tag{21}$$

At this point, from Eq. (17),
$$c_\oplus^0 \exp\left(-z_\oplus u(b)\right) = c_\ominus^0 \exp\left(-z_\ominus u(b)\right).$$

Taking the natural logarithm,
$$-z_\oplus u(b) + \ln c_\oplus^0 = -z_\ominus u(b) + \ln c_\ominus^0,$$

which simplifies to
$$(z_\ominus - z_\oplus)u(b) = \ln\left(\frac{c_\oplus^0}{c_\ominus^0}\right).$$

Since $c_\oplus^0 = c_\ominus^0$ and $z_\ominus \neq z_\oplus$, it follows that $u(b) = 0$, so $\phi(b) = 0$. Therefore, at $x = b$, the total potential $\phi(b)$ is zero, and the internal potential satisfies $\phi_{\text{int}}(b) = -\phi_{\text{ext}}(b)$.

To relate the total potential drop to the integral over ions, consider the total electric field (13). Integrating as
$$\phi(x) = \phi(0) - \int_0^x E(y)\,dy = \phi(0) + \int_0^x \left(\frac{d\phi_{\text{int}}}{dy} + \frac{d\phi_{\text{ext}}}{dy}\right)dy. \tag{22}$$

Since $\phi_{\text{ext}}(x)$ is known and $\phi_{\text{int}}(x)$ can be obtained from Poisson's equation, this expression allows us to calculate $\phi(x)$. Specifically, integrating Poisson's equation Eq. (18), we have
$$\phi_{\text{int}}(x) = -\int_0^x \int_0^y \frac{\rho(z)}{\varepsilon_0}\,dz\,dy.$$

Therefore, the total potential is
$$\phi(x) = \phi(0) + \int_0^x \frac{d\phi_{\text{ext}}}{dy}\,dy - \int_0^x \int_0^y \frac{\rho(z)}{\varepsilon_0}\,dz\,dy.$$

This shows that the total potential drop includes contributions from both the external potential and the internal potential due to ion distributions, relate the total potential and ion concentrations, showing how the potential drop is connected to the distribution of ions.

Under steady-state conditions, the ionic flux is zero, meaning diffusion and electromigration reach equilibrium. From the Nernst-Planck equation under steady-state conditions, Eq. (15), leads to Eq. (16), which we have already integrated to obtain Eq. (17). Substituting $c_i(x)$ into Poisson's equation for the internal potential (19), near $x = b$, due to charge neutrality, satisfied (21). Solving for $\phi(b)$ establishes the relationship between the potential difference and ion concentrations:
$$\phi(0) - \phi(b) \approx \frac{k_B T}{z_\oplus e} \ln\left(\frac{c_\oplus^0}{c_\oplus(b)}\right),$$
$$\phi(b) - \phi(l) \approx \frac{k_B T}{z_\ominus e} \ln\left(\frac{c_\ominus^0}{c_\ominus(l)}\right).$$

## H   GRADIENT BACKWARD FLOW

Gradient backward flow refers to how gradients propagate through a neural network during backpropagation, especially in multimodal fusion settings. Understanding this flow is essential for addressing gradient conflicts and enhancing model performance.

## H.1 Definition and Explanation

In backpropagation, gradients flow from the output back to the input, adjusting parameters based on the downstream task loss. As illustrated in Figure 9, the gradient from the downstream task loss $\mathcal{L}_{\text{task}}$ propagates through the entire network, while the gradient from the fusion stage loss $\mathcal{L}_{\text{fuse}}$ (if present) propagates from the fusion output towards the downstream task. Consequently, the feature extractor's parameters are influenced by gradients from both losses. However, gradients from $\mathcal{L}_{\text{task}}$ and $\mathcal{L}_{\text{fuse}}$ may conflict, leading to gradient interference. Balancing these conflicting gradients often requires hyperparameters to weigh the influence of different losses. For example, in variational autoencoders (Kingma & Welling, 2014), the Kullback-Leibler divergence loss promotes generalization, while the reconstruction loss focuses on accurate reconstruction. Adjusting their relative weights is crucial yet challenging.

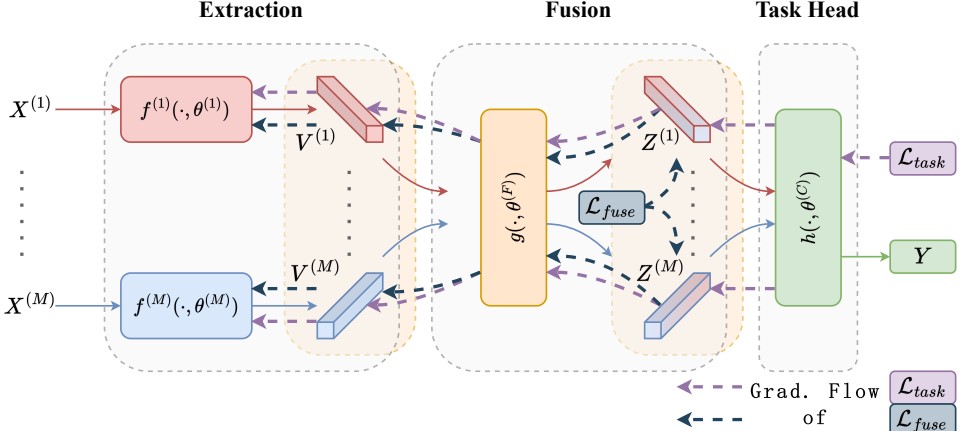

Figure 9: Gradient flow diagram extended from Figure 6. The notation is consistent with Figure 6. Dark blue dotted arrows represent gradients from the fusion stage loss ($\mathcal{L}_{\text{fuse}}$), light purple dotted arrows represent gradients from the downstream task loss ($\mathcal{L}_{\text{task}}$).

In multimodal tasks, similar conflicts arise. Fusion-related losses, such as contrastive loss (Radford et al., 2021), aim to align features from different modalities. However, enforcing such alignment may not always be beneficial, especially when modalities carry distinct and complementary information (Liang et al., 2022; Jiang et al., 2023). Overemphasis on aligning modalities can lead to the loss of modality-specific information, degrading overall performance, as discussed in Sec. 5. Introducing fusion losses directly affects feature extractors by imposing prior assumptions on the extracted features, potentially leading to suboptimal representations and increased computational cost.

## H.2 Theoretical Analysis with Residual Connections

Residual networks (He et al., 2016) address degradation in deep networks by introducing skip connections, which alleviate gradient vanishing and exploding issues, as shown in Figure 10. The residual connection introduces an identity mapping that facilitates gradient flow, providing a direct path for gradients and ensuring effective training of deeper networks.

In multimodal fusion networks, similar gradient issues can arise due to deep fusion mechanisms introducing additional layers between feature extractors and downstream tasks. These layers can cause gradients to diminish or explode, leading to ineffective training of feature extractors. Applying residual connections in the fusion stage can alleviate these issues by providing direct gradient pathways (Shankar et al., 2022; Wang et al., 2023a). However, residual connections require strict dimension consistency between inputs and outputs, which may not always be feasible in multimodal settings where modalities have different dimensions. Moreover, gradient interference cannot be entirely ruled out since gradients still merge in the extraction phase.

An alternative is to use concatenation-based approaches similar to DenseNet (Huang et al., 2017), where features are stacked along the channel dimension. While this avoids some issues of residual

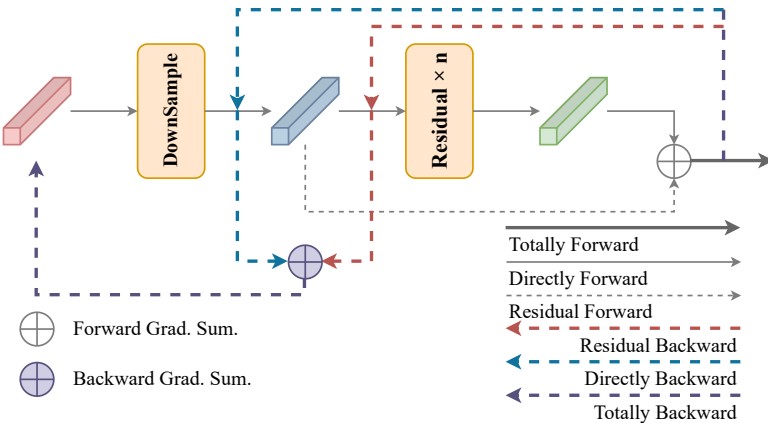

Figure 10: Structure of a residual block in networks.

addition, it can increase feature dimensions, causing redundancy and potential overfitting. Therefore, while residual connections can help address gradient conflicts in deep networks, they are not a complete solution for conflicts arising from conflicting loss functions in multimodal fusion. Limiting the depth of the fusion network or carefully designing fusion loss functions to align with downstream task objectives can help mitigate gradient conflicts and improve overall performance.

# I  TRI-MODAL EXPERIMENTS

Tab. 10 reports accuracy and F1 on CMU-MOSEI Bagher Zadeh et al. (2018) under the fixed missing protocol Zhao et al. (2025), where each model is evaluated on all seven modality combinations $\{a\}$, $\{t\}$, $\{v\}$, $\{a, v\}$, $\{a, t\}$, $\{t, v\}$ and $\{a, t, v\}$. Consistent with prior work, text-only performance is already very strong, while acoustic-only and visual-only conditions are substantially harder. This reflects the text-dominant nature of CMU-MOSEI, where most sentiment cues can be inferred from transcripts and acoustic/visual streams mainly provide complementary information in challenging cases.

Existing methods already show a clear progression along this metric: early fusion-based models such as MCTN, MMIN and GCNet provide a reasonable starting point, later incomplete-modality methods (IMDer, DiCMoR, MoMKE, EUAR) yield consistent gains, and the recent MCULoRA achieves the strongest overall performance, particularly on text-dominated and full-modality settings. This trend establishes MCULoRA as a competitive reference point and highlights that further improvements must come from more robust handling of weak and missing modalities rather than from simple architectural scaling.

We first compare our plain feed-forward fusion variant (FFN) with existing non-PMR baselines. FFN is deliberately minimalist: it replaces the PMR block with a shallow MLP while keeping the encoders and training protocol identical. As shown in the table, FFN achieves performance comparable to GCNet on most modality combinations and slightly improves over earlier architectures such as MCTN and MMIN. This indicates that our backbone and training setup already form a strong baseline, and any additional gains from PMR cannot be attributed to trivial re-implementation details.

The rG-SRHT variant applies a parameter-free subsampled randomized Hadamard transform inside the PMR block. This structured random projection tends to smooth spurious spikes in the acoustic and visual representations and approximately preserves their energy. As a result, rG-SRHT achieves competitive or slightly better performance than EUAR under purely weak-modality conditions ($\{a\}$, $\{v\}$, and $\{a, v\}$), where robustness to noisy low-signal inputs is crucial. However, because SRHT is not learnable, it also flattens discriminative directions in the text representation. On text-dominated combinations ($\{t\}$, $\{a, t\}$, $\{t, v\}$, $\{a, t, v\}$), rG-SRHT is consistently but modestly below EUAR, leading to a slightly worse overall average. This behavior is consistent with the intuition that random orthogonal transforms help regularize weak modalities but cannot fully exploit the rich structure of the textual features.

Table 10: Accuracy comparison under fixed missing protocol on CMU-MOSEI.

| Datasets | Models | Testing condition (ACC/F1, %) | | | | | | | $\{a,t,v\}$ |
|---|---|---|---|---|---|---|---|---|---|
| | | $\{a\}$ | $\{t\}$ | $\{v\}$ | $\{a,v\}$ | $\{a,t\}$ | $\{t,v\}$ | Avg. | |
| CMU-MOSEI | MCTN Pham et al. (2019) | 62.7/54.5 | 82.6/82.8 | 62.6/57.1 | 63.7/62.7 | 83.5/83.3 | 83.2/83.2 | 73.1/70.6 | 84.2/84.2 |
| | MMIN Zhao et al. (2021) | 58.9/59.5 | 82.3/82.4 | 59.3/60.0 | 63.5/61.9 | 83.7/83.3 | 83.8/83.4 | 71.9/71.8 | 84.3/84.2 |
| | GCNet Lian et al. (2023) | 60.2/60.3 | 83.0/83.2 | 61.9/61.6 | 64.1/57.2 | 84.3/84.4 | 84.3/84.4 | 73.1/72.8 | 85.2/85.1 |
| | IMDer Wang et al. (2023c) | 63.8/60.6 | 84.5/84.5 | 63.9/63.6 | 64.9/63.5 | 85.1/85.1 | 85.0/85.0 | 76.0/75.3 | 85.1/85.1 |
| | DiCMoR Wang et al. (2023b) | 62.9/60.4 | 84.3/84.4 | 63.6/63.6 | 65.2/64.4 | 85.0/84.9 | 85.0/84.9 | 75.9/75.4 | 85.1/85.1 |
| | MoMKE Xu et al. (2024) | 65.9/65.5 | 80.8/80.7 | 64.9/64.8 | 65.9/65.5 | 86.0/85.9 | 84.4/84.3 | 76.4/76.2 | 86.7/86.6 |
| | EUAR Gao et al. (2024) | 64.5/60.7 | 85.3/85.2 | 66.4/65.3 | 66.5/65.4 | 85.6/85.1 | 86.0/86.0 | 77.3/76.3 | 86.6/86.4 |
| | MCULoRA Zhao et al. (2025) | 68.5/70.2 | 86.9/87.0 | 69.6/71.1 | 71.0/72.4 | 87.2/87.3 | 87.0/87.1 | 79.6/80.3 | 87.2/87.3 |
| | **FFN** | 60.0/60.0 | 82.9/83.1 | 61.7/61.5 | 64.0/57.3 | 83.7/83.9 | 83.6/83.7 | 74.4/73.4 | 84.6/84.6 |
| | **rG-SRHT** | 65.0/61.3 | 85.0/84.9 | 66.1/65.7 | 66.4/65.2 | 85.3/84.9 | 85.8/85.4 | 77.1/76.2 | 86.3/86.1 |
| | **rG-CMMP** | 69.0/70.6 | 86.2/86.3 | 70.1/71.5 | 71.6/72.9 | 86.7/86.9 | 86.3/86.7 | 79.5/80.3 | 86.8/86.9 |
| | **rG-UMoE** | 69.3/71.0 | 87.3/87.4 | 70.4/72.0 | 72.0/73.3 | 87.7/87.9 | 87.6/87.8 | 80.3/81.1 | 87.8/88.0 |

In contrast, rG-CMMP couples PMR with a contrastive multi-modal projection head. By jointly enforcing semantic alignment and information preservation in the shared space, rG-CMMP improves robustness when text is absent. Under $\{a\}$, $\{v\}$, and $\{a,v\}$, it outperforms MCULoRA, suggesting that PMR helps the acoustic and visual streams retain task-relevant cues even in the presence of large modality gaps. Nevertheless, in text-heavy conditions, rG-CMMP is slightly weaker than MCULoRA. The contrastive objective pulls the text representation towards a compromise that better matches the weaker modalities, which improves cross-modal consistency but marginally degrades the fine-grained decision boundary that text alone could achieve. Consequently, the overall average and the full-modality $\{a,t,v\}$ score of rG-CMMP remain close to but slightly below those of MCULoRA.

Finally, rG-UMoE augments rG-CMMP with a sparse unified mixture-of-experts layer. The expert routing allows the model to specialize different subspaces for text-dominated and weak-modality-dominated inputs, thereby mitigating the over-regularization of the text encoder while preserving the benefits of PMR and contrastive alignment. Empirically, rG-UMoE consistently matches or surpasses MCULoRA across all modality combinations, with the most notable gains on the more challenging weak-modality and mixed-modality settings. On the full $\{a,t,v\}$ configuration and in terms of the Average metric, rG-UMoE achieves the best performance among all compared methods. These results demonstrate that injecting a lightweight expert structure into the PMR block is an effective way to reconcile strong text performance with robustness to missing and noisy modalities on CMU-MOSEI.

