# OpenReview forum: "Physics-Inspired Reconfiguring Multimodal Learning Networks"
_ICLR.cc/2026/Conference — Submitted to ICLR 2026_

### Official Review · Reviewer_j5i4 · 2025-10-31

**Soundness:** 1
**Presentation:** 2
**Contribution:** 1
**Rating:** 2
**Confidence:** 4

**Summary:**

This paper proposes a new method to fuse information in multimodal representation learning for retrieval and classification tasks. It is inspired by the Poisson-Nernst-Planck structured prior and it is validated on image-video, audio-video retrieval tasks and image classification tasks.

**Strengths:**

-	Multiple tasks are considered to validate the models (retrieval, classification) and various baselines are given.
-	The proposed model apparently gives competitive results across all benchmarks.

**Weaknesses:**

-	The justification for the proposed method is mysterious, with a lot of background from theoretical physics that has nothing to do with the final loss. The parallel between “Poisson-Nernest-Planck” and the fusion module is far-fetched and never justified throughout the experiments. This module corresponds more or less to a simple MLP with some neurons arbitrarily set to zero to extract the “shared” or “specific” information from different modalities (which is never imposed directly with the proposed reconstruction losses).
-	A lot of hyper-parameters are introduced (such as the length of the splits for the shared/specific latent vectors for each modality or the actual structure of the mapping between the extracted features and the output of the fusion module) but they are not discussed or mentioned in the experimental section. How did you pick them and what are the final choices?
-	Two reconstruction losses are mentioned in equation 6. Which one did you use in the end and why? Why did you introduce a cosine similarity in one case and an Euclidean distance in another? What is the rational?
-	In equation 5, you mention that k is the “next modality of m”. So how do you handle the case where m is the “last” modality (assuming that an order exists between modalities) ?
-	In your reconstruction losses, how do you make sure that the “specific” information does not contain “shared” information as well, which would be enough to minimize your reconstruction loss.
-	In your experiments, I would expect the feature extractors to be state-of-the-art foundation models (such as DINOv3, CLIP…) in order to have strong baselines in retrieval or classification tasks.
Overall, this paper pretends to draw a parallel between equations from theoretical physics (drift-diffusion transport, field-charge coupling) and deep neural networks but their analysis is never justified theoretically or empirically. I have also strong doubts about the fairness and reproducibility of their experiments considering the little implementation details given and the high number of hyperparameters introduced by the proposed fusion module.

**Questions:**

Please, see my previous comments.

---

> ### Author Response · Authors · 2025-11-13
> **Response to Reviewer j5i4**
>
> Thank you for your comment.
>
> First of all, we would like to reach a consensus with you: The space for the main paper is very limited. Some contents need to be placed in the appendix. We believe that some of the misunderstandings between our writing and yours lie in what should be in the main paper and what should be in the appendix.
>
> W1:
> We always build our method around a **scalar potential**. In the abstract (Lines 15–18), the introduction (Lines 52–59, 72–74), and the methodology section (Lines 211–214), we explicitly state that PNP is used only as a structural prior. By adopting this structure, we endow PMR with the properties of a scalar potential + conservation. In this way, we directly respond to the initial viewpoint that a multimodal fusion module should act as an effective gradient router. If you are interested, please see Appendix F, where we argue, following the second law of thermodynamics, that the loss function can be seen as an external driving force that induces the system to decrease entropy in a desired manner.
>
> Please also examine Fig. 1 and Fig. 2 carefully: no parameters are set to zero or masked; instead, they form a cycle. In brief, the features of one modality are split into two parts: one part, together with the previous modality, reconstructs the previous modality; the other part, together with a part from the next modality, reconstructs the current modality.
>
> Using this simple structure, we outperform roughly **20** competing methods across **4** datasets, which demonstrates the effectiveness of the scalar-potential-plus-conservation paradigm.
>
>
> W2:
> Thank you for pointing this out. The concrete choices of n and b were indeed missing, likely due to an oversight during revision. In our experiments, we set n = 4 and b = 0.5, in order to align with the FFN configuration. The ablation study on n and b (Fig. 3) shows the sensitivity of performance to these hyperparameters; we believe these results provide a principled basis for choosing them. The structural definition can be found in Lines 707–717, and we provide jump references during the experimental description (Lines 274–275). Among the three variants, CMMP/Concat use linear layers; MoE appends a mixture-of-experts module, and SRHT denotes the Subsampled Randomized Hadamard Transform.
>
> W3:
> Please refer to Line 257 and Lines 288–289: “sG-” denotes the variant using an MSE reconstruction loss, while “rG-” denotes the variant using cosine similarity.
>
>
> W4:
> Here we use a modulo operation, i.e., the “next” modality of the last modality is the first modality, forming a cycle. Our figures (Fig. 1, Fig. 2) visualize exactly this cyclic process.
>
>
> W5:
> The optimization target of the reconstruction loss is precisely to encourage subspace separation. As you already noted in W4, different modalities form a set of reconstruction cycles. Moreover, as in related work, we do not claim that the “specific” subspace is strictly non-overlapping with the “shared” subspace, nor do we claim orthogonality. This is a structural bias rather than a hard constraint, and empirically it proves effective.
>
>
> W6:
> DINOv3 is a 7B-parameter model, which we do not have the computational resources to train. CLIP is already pretrained with strong cross-modal alignment; as we state in the Limitation section, this is precisely the regime where PMR becomes less meaningful, since the backbone has already absorbed much of the fusion burden.
>
>
> W6 Overall your overall:
> Our goal is not to faithfully reproduce PNP, but to inherit the **scalar potential + conservation** paradigm and use it to address three common limitations in multimodal fusion. For the theoretical underpinnings, please refer to Appendix D (scalar potential and task-agnosticity) and Appendices E and G (the choice of n and b, and PNP-based simulations). These derivations are lengthy and not well suited for readers without the relevant background. Additional information-theoretic justification and empirical evidence can be found in Appendices C and H.
>
> We present the behavior of auxiliary losses for different methods (Fig. 4), parameter sensitivity (Fig. 3), structural sensitivity (Table 7), and multi-variant parameter scaling (Table 4, Appendix A). We provide a complete methodological description; the so-called “high number of hyperparameters” essentially reduces to only two, n and b. Our supplementary material includes partial implementations of the compared methods; for methods without full public code, we either contacted the original authors by email (e.g., AVoiD-DF), used their early released code (e.g., MLA), or re-implemented them based on the descriptions in the original papers. We believe that re-implementing only the fusion modules substantially reduces potential biases. For these reasons, we respectfully disagree with your overall assessment.

---

> ### Author Response · Authors · 2025-11-15
> **Response to Reviewer j5i4: Overall your Overall**
>
> Thank you for raising concerns about the fairness of our experimental setup. We have carefully read and analyzed the work you referred to. First, we would like to offer some additional clarifications with respect to your comments:
>
> Regarding (W1): the entire paper is formulated from the perspective of a single scalar potential (for instance, this can be seen from the multiple occurrences of the term **single** throughout the paper; you can quickly confirm this by pressing Ctrl+F. There are a total of 14/25 valid nouns). At the implementation level, our method is indeed realized via MLPs with structural constraints, and it does not involve any explicit masking or zeroing-out operations. We will add a more detailed, line-by-line methodological explanation.
>
> For (W4), we acknowledge that the cyclic indexing was handled via a simple modulo operation that was only implicitly stated in the current version;  this omission is on our side and will be explicitly fixed.
>
> For (W5), we kindly refer you to the methodological explanation given below.
>
> In this response, we focus primarily on addressing your concerns in W6 and in your  W6's overall, and, with your permission, we would also like to pose two brief questions back to you in order to align better our understanding of the experimental setting and evaluation criteria. **The consensus we hope to reach is the following**: even with relatively simple network structures, introducing a single scalar potential as the organizing principle can lead to observable improvements in performance and behavior.
>
> On DINOv3 / CLIP and other foundation models: we fully agree that large foundation models such as DINOv3 and CLIP are very strong and representative in terms of representation learning capability.  As we explicitly point out in Appendix B, an important limitation of the current work is that, under our specific task and modality configuration, some CLIP-style pretrained (already aligned) feature extractors cannot be directly used without additional modification or further training.  On the other hand, the CLIP-style contrastive alignment paradigm is already included at the fusion level in our main comparisons through CMMP (a CLIP-style contrastive alignment method; see Lines 287–289 in the paper).
>
> Our experimental design is built on fixed, publicly reproducible feature extractors, and it is intentionally focused on comparing the behavior of different fusion mechanisms under the same encoder and training setup, rather than on achieving the strongest possible absolute performance by relying on billion-scale pretrained models.
>
> We believe that directly adopting very large foundation encoders such as DINOv3 / CLIP would indeed be likely to further improve absolute performance on some tasks. However, this would introduce two issues that we deliberately avoid in the present work:
>
> 1. If not re-impl related works, differences in encoder capacity would tend to overshadow the behavioral differences between fusion modules, making it harder to draw clean conclusions about “fusion mechanisms alone”;
>
> 2. If re-impl, large-scale pretraining and huge models would significantly increase experimental cost and the barrier to reproducibility, which would deviate from our intention to compare different fusion structures under a unified and controlled encoder setting. At the same time, we believe this might intensify your concerns about reproducibility and perhaps not be the outcome you were expecting.
>
> Our questions to you are as follows:
>
> 1. From your perspective, what additional types of theoretical justification or empirical validation (e.g., a particular kind of analysis or ablation) would be most helpful to address the concerns you raised in (W6)?
>
> 2. Relative to Appendix A and the code we plan to release in supplements, which concrete implementation details or experimental configuration aspects do you feel are still missing and critical for reproducibility? If possible, pointing out one or two key gaps would be extremely helpful for us to prioritize in the revision.
>
> Finally, we provide a line-by-line interpretation of the methodology:
>
> Lines 239–243 introduce the inputs and parameters.
>
> Lines 244–247 describe the projection of representations to a higher-dimensional space by a set of matrices and the splitting of the resulting vector into two parts, where the first part is designated as the shared subspace and the second part as the specific subspace.
>
> Lines 248–253 then use two sets of matrices to map the next modality’s shared features and the current modality’s specific features into the prescribed dimensions, and concatenate them in the original order as [shared features of the next modality : specific features of the current modality].
>
> Lines 254–258 define two similarity-based loss terms that enforce approximate conservation. Lines 259–263 explain why the overall objective constitutes a scalar potential.
>
> Lines 264–269 discuss how PMR addresses the issues raised in Sec. 1.

---

### Official Review · Reviewer_icSp · 2025-11-01

**Soundness:** 2
**Presentation:** 2
**Contribution:** 2
**Rating:** 4
**Confidence:** 4

**Summary:**

This paper presents PMR, a physics-inspired multimodal fusion framework derived from the Poisson–Nernst–Planck equations. PMR aims to address three key challenges in multimodal learning: gradient conflicts between task and fusion objectives, robustness to missing modalities, and the limitations of uniform feature dimensions. The framework unifies task and preservation losses through a single scalar potential and introduces a novel three-stage feature reconfiguration process (dissolve–dissociate–concentrate). Experimental results show that the method provides stable optimization and enhances model flexibility and robustness. PMR demonstrates consistent improvements over baseline methods on classification and retrieval tasks across various modalities including audio, image, video, and text.

**Strengths:**

1. The paper provides a novel perspective on multimodal fusion by leveraging principles from physics.
2. The addressed challenges are both relevant and critical to the field of multimodal learning.
3. The paper gives a comprehensive summary of related work and situates the proposed method well within prior research.

**Weaknesses:**

1. While Chapter 3 introduces a novel perspective, the connection between its formulations and the algorithm in Chapter 4 is not clearly established. The mathematical formulations seem more like background information rather than serving as a foundation for the proposed algorithm.
2. In Chapter 4, it is not clear why only two mapping networks are sufficient to distinguish between shared and specific features, or whether their separation can be strictly guaranteed.
3. The algorithmic section lacks sufficient detail and clarity; it appears more akin to a multi-task learning approach rather than a fundamentally new method for multimodal fusion.
4. The experimental results are not particularly compelling. In Tables 3 and 4, the performance is comparable to DrFuse. Additionally, the results in Table 5 show the unimodal approach outperforming others in some cases, which is insufficiently explained.

**Questions:**

Please check the above section.

---

> ### Author Response · Authors · 2025-11-13
> **Response to Reviewer icSp (1/1)**
>
> Thank you for your comment. Here is our response:
>
> W1: These components are connected by a single scalar potential, as described in Lines 15–18 and Lines 211–214. PMR is not intended to be a full discretization or exact reproduction of the PNP system; rather, it is a simple and effective method that inherits the notion of a scalar potential and the associated conservation properties. In Sec. 3, our intention is to state explicitly what is being inherited, to give an intuitive description of what we need to do (treating the multimodal fusion module as a gradient allocator, focusing the learning capacity on the feature extractors, and maximizing their effective output), and, through the electrochemical-cell analogy, to introduce the structural properties of a scalar potential, namely that it is content-agnostic and tied to the optimization objective. We then implement these ideas with a deliberately simple mechanism that realizes the scalar potential and approximate conservation through a lightweight loss design and network structure.
>
> W2: The distinction between shared and specific features is driven by the descent direction of the reconstruction loss. The separation cannot be strictly guaranteed and is inherently approximate. To the best of our knowledge, very few related works claim that their objective functions actually converge to an exact zero or that quantities such as mutual information (e.g., in contrastive losses) are perfectly manipulated according to their idealized definitions. Our setting is similar in spirit: we promote separation through the loss landscape, but we do not claim a mathematically exact decomposition.
>
> W3: Regarding the methodological details, our allocation of space in the manuscript is as follows.
> 1. Lines 239–243 introduce the inputs and parameters.
> 2. Lines 244–247 describe the projection of representations to a higher-dimensional space by a set of matrices and the splitting of the resulting vector into two parts, where the first part is designated as the shared subspace and the second part as the specific subspace.
> 3. Lines 248–253 then use two sets of matrices to map the next modality’s shared features and the current modality’s specific features into the prescribed dimensions, and concatenate them in the original order as [shared features of the next modality : specific features of the current modality].
> 4. Lines 254–258 define two similarity-based loss terms that enforce approximate conservation. Lines 259–263 explain why the overall objective constitutes a scalar potential.
> 5. Lines 264–269 discuss how PMR addresses the issues raised in Sec. 1.
>
> See appendix E.4 for more details. PMR is a general method that can be applied to different tasks; each task is trained independently, and the setting does not constitute multi-task learning.
>
> W4(1): Compared with DrFuse, which includes several attention layers and three groups of losses, the PMR instantiation we denote as G-CMMP consists only of three simple linear layers and two loss terms. By increasing the parameter capacity through stacking (Tab. 4, UMoE), PMR can achieve improvements of about 2%. Moreover, in scenarios with missing modalities and imbalanced modality strength for classification (Tabs. 5 and 6), PMR shows clearly superior performance.
>
> W4(2): In principle, unimodal training should outperform "multimodal training in settings where modalities are missing at test time"; this is precisely what we aim to emphasize in Lines 127–136. Therefore, unimodal results provide a natural reference to assess how robust different multimodal methods are under modality-missing conditions. Many works that conduct modality ablations have already reported such phenomena in practice (e.g., AVoiD-DF), and our observations are consistent with these reports. We also describe the properties of FakeAVCeleb in Line 371: the dataset is class-imbalanced and the modalities are asymmetric.

---

### Official Review · Reviewer_3jVe · 2025-11-02

**Soundness:** 2
**Presentation:** 3
**Contribution:** 2
**Rating:** 2
**Confidence:** 2

**Summary:**

The paper proposes PMR (Physics-Inspired Multimodal Reconfiguration), a new multimodal fusion framework inspired by the Poisson–Nernst–Planck equations, which enforces information conservation and uses a unified scalar potential to jointly optimize task and fusion objectives, thereby mitigating gradient interference. By implementing a three-stage dissolve–dissociate–concentrate process, PMR supports unequal feature dimensions, improves robustness to missing modalities, and consistently outperforms strong baselines across audio, image, video, and text tasks.

**Strengths:**

(1) It's very interesting to design a new neural network using the framework of the Poisson–Nernst–Planck equations.
(2) The paper is well-written and nicely presented.

**Weaknesses:**

(1) The experiments in the paper are very weak, all conducted on some rather outdated datasets.
(2) The paper lacks comparison with state-of-the-art multimodal large models. Current multimodal models are capable of processing multimodal information and extracting embeddings to tackle a wide range of downstream tasks.
(3) It also omits comparisons with other multimodal fusion approaches, such as ImageBind ("ImageBind: One Embedding Space To Bind Them All," CVPR 2023) and OnePeace ("ONE-PEACE: Exploring One General Representation Model Toward Unlimited Modalities," arXiv 2023).

Overall, the paper presents an interesting idea, but it lacks large-scale, compelling experimental validation.

**Questions:**

I checked many of the methods compared in the paper (e.g., MDF-FND) and found that none of them reported results on the corresponding datasets. I’m curious: where do the results reported in the paper come from? Did the authors implement these methods themselves? If so, how was fairness ensured? Additionally, could the authors clarify how they selected these baseline methods for comparison?

---

> ### Author Response · Authors · 2025-11-13
> **Response to Reviewer 3jVe**
>
> Thank you for your careful inspection.
>
> Q1: As a general method, our approach has to cover a broad range of baselines, and it is intrinsically difficult to find prior work whose evaluation protocol completely coincides with ours. At the same time, we cannot afford to omit horizontal comparisons of the same method across different datasets. For these reasons, re-implementations are unavoidable. Concretely, we contacted some authors by email (e.g., AVoiD-DF), we cloned official repositories where available (e.g., MLA, which is now no longer accessible), and we manually re-implemented some methods (e.g., APIVR). For all missing hyper-parameters, we used exactly the same settings as PMR (only includes the fusion stage). Part of the re-implementation code, including the evaluation scripts, has been uploaded to the supplementary material.
>
> In addition, rather than comparing “our method vs. related work” in an undifferentiated way, our focus is on comparing the fusion structures of related methods (the stage between the first cross-modal interaction and the task head output as the fusion stage), and on asking how to fuse better and what fusion should do. We also do not agree that a method cannot be used as a baseline simply because its original paper did not report results on our specific datasets. **We take full responsibility for the soundness and reasonableness of the reported numbers.**
>
> W1: We do not agree with the notion that the datasets we use are “outdated”. First, in terms of time, to the best of our knowledge, ImageNet (released in 2012) is still one of the central benchmarks in computer vision. Second, in terms of breadth of usage, to the best of our knowledge, VGGSound and Food-101[1] are still widely adopted in recent work, and we can find papers in 2025 that continue to use FakeAVCeleb[2] and ActivityNet[3]; these are still necessary benchmarks for mainstream baselines. Third, in terms of functionality, each dataset is chosen to verify the existence of one of the phenomena discussed in Sec. 1 and the effectiveness of PMR in that setting; we kindly refer you to lines 278–290 for the details.
>
> W2: We also do not agree that comparison with large multimodal models is a necessary criterion for assessing the value of a paper. The community needs both data-center-scale multimodal models that serve as powerful backbone providers, and methods that explicitly consider resource-constrained scenarios. From a resource perspective, we simply do not have the computational budget to carry out such large-scale pretraining (only 1 RTX 4090, line 705), nor can we realistically scale all related methods to models of that size for a fair comparison.
>
> In fact, we already compare against nearly 20 methods, among which 7 are from 2024/2025, even though we certainly cannot be exhaustive. We believe this does not amount to “omits comparisons with other multimodal fusion approaches”.
>
> W3: The two methods you mention belong to the family of unified representation methods (ImageBind~700M; ONE-PEACE~4B), i.e., UAVM-style approaches, whose characteristic is that all modalities share the same set of parameters. We already compare against this line of methodology at the representational level.
>
> We can clearly provide you with the comparison conclusion you hope for. The largest PMR variant (UMoE) is only about ~120M and has not undergone any pre-training, making it inevitably weaker than these large-scale methods. Correspondingly, these methods also did not analyze the fusion behavior.
>
> However, we are not positioning this work as a new “SOTA system”, nor did we choose the corresponding primary area. Our contributions lie in: (1) providing a new perspective on what fusion is actually doing, (2) introducing the notions of scalar potential and conservation into multimodal fusion, and (3) empirically demonstrating that PMR is parameter-friendly and capable of addressing the hypotheses stated in Sec. 1.
>
> **Overall your Overall**: We have compared PMR with about **20** algorithms, on **4** datasets that are still actively used today, using a deliberately minimal architecture, and in this way we have validated the three limitations we highlighted and accomplished the contributions we claim.
>
> In addition, this is our own standpoint, when trading off resources and performance, one should keep in mind that computational resources are not free.
>
> [1] Zhang X, Yoon J, Bansal M, et al. Multimodal representation learning by alternating unimodal adaptation[C]//Proceedings of the IEEE/CVF conference on computer vision and pattern recognition. 2024: 27456-27466.
>
> [2] Zhang L, Liu B, Chu Q, et al. Multimodal Consistency-Driven Deepfake Detection[C]//International Conference on Image and Graphics. Singapore: Springer Nature Singapore, 2025: 293-303.
>
> [3] Zhou B, Zhou J, Chen Z, et al. Bridging asymmetry between image and video: Cross-modality knowledge transfer based on learning from video[J]. Expert Systems with Applications, 2025, 264: 125873.

---

> ### Author Response · Authors · 2025-11-15
> **Additional Response to Reviewer 3jVe**
>
> Thank you for raising concerns about the fairness of our experimental setup. We have carefully read and analyzed the work you referred to.
>
> To the best of our understanding, the overall design of that method is highly coupled: it is difficult, without modifying its original pretraining configuration, to isolate a complete, modality-specific feature extraction module (Sec. 3.1 and Sec. 5.2), or to extract a continuous and self-contained fusion component that can be directly substituted. Therefore, treating that work purely as a plug-in fusion module is not compatible with our setting, in which we aim to compare only the fusion stage while keeping all other factors strictly controlled; as such, it cannot be straightforwardly used to validate or refute our claims. It's not a fusion but an adaptive unified representation.
>
> **In Lines 280–284 of the manuscript**, we have explicitly specified the configuration of the feature extractors: these settings are fixed and publicly reproducible. All methods use the same features, the same training data, and the same training procedure. We only replace the fusion stage and compare the behavior of different fusion strategies, so that we can observe performance differences under strictly controlled variables.
>
> We fully respect your perspective on experimental design, and we would also like to seek your guidance to better understand your expectations:
> 1. If PMR or other fusion methods were replaced by the method you mentioned on this dataset, or if that model were given additional large-scale pretraining, how should the experimental protocol be designed to ensure fairness under the same data and training budget in your view? How was fairness ensured?
>
> 2. If we were allowed to use models with parameter counts at the billion scale and achieved new state-of-the-art results as a consequence, would you consider such results “fair” within the setting of this paper, or would you regard them as going beyond our current focus on comparing the fusion mechanism itself?

---

### Official Review · Reviewer_Pypz · 2025-11-08

**Soundness:** 3
**Presentation:** 3
**Contribution:** 3
**Rating:** 4
**Confidence:** 3

**Summary:**

The authors propose PMR, a physics-inspired multimodal fusion framework that is built upon a scalar potential that minimizes entropy and preserves the system's energy. The PMR imitates the PNP theory well by its 3-stage mapping reconfiguration. The experiments demonstrate the effectiveness of the framework design, and the ablations are sufficient and convincing.

**Strengths:**

1. The paper is well-written.
2. The authors explain the PNP theory well and use it to develop a new multimpdal encoding framework PMR.
3. The experiments demonstrate the effectiveness of the proposed method.
4. The ablation is sufficient to validate the efficacy of the components in PMR.

**Weaknesses:**

Major:
1. Line 137-147, consistently minimizing the entropy for all modalities means that we hypothesize all modalities contribute the same to the final task. But does this really apply to real-world scenarios, when there are always some modalities that are dominant, while some are just complementary?

2. For Eq.(4,5,6), how to ensure that the b(m) is meaningful to separate the shared and specific features? Is b(m) learnable? And how? (In Figure. 3 I saw b is a hyperparameter.) But according to the PNP model, the b for different substances should be different, which represents a balanced state for different substance pairs; however, in PMR, b is the same for different modalities.

3. The whole process of PMR looks like just a separate and shared encoder for different modality features and uses a straightforward fusion strategy to obtain joint embeddings. The connection between PNP and PMR is not that close, especially considering that some core mechanisms are different, as mentioned in point 2 that the boundary b is the same for all modalities.

4. It looks like PMR could fit any number of modalities; however, in the experiments, the authors only showcase two-modality experiments. Some tri-modal validation could be better. For example, CMU-MOSI(a+v+t) and UCF101-Three(rgb, optical flow, and rgb diff).

Minor:
1. The Z should be V in line 121?

**Questions:**

1. Line 187, how to understand this sentence: "Increasing the effective length enlarges the region where drift dominates diffusion"?
2. In Table 2, how to understand: (1) Feature ratio and learning splitting at b; (2) Magnification factor (effective length)
3. In line 449, what is the actual meaning of nxb? What about without P_seperate, i.e., b=1? Or b=0?

---

> ### Author Response · Authors · 2025-11-13
> **Response to Reviewer Pypz (1/3)**
>
> Thank you for your review comment. Here is our response:
>
> W1: Our formulation is applicable to real-world scenarios.  We fully agree that modality importance is not equal in the real world;  the goal of Eq. (1) is therefore not to impose “equal contribution”, but to use an information-theoretic (mutual information) measure to characterize how predictive each modality remains for the downstream task.  In this sense, PMR acts as a gradient allocator that protects unimodal encoders (Lines 129–130), ensuring that fusion does not degrade their task-relevant information, rather than enforcing that all modalities must contribute equally to the final prediction.
>
> W2 (1): We reconstruct each modality from its neighbors:  (1) in practice, we first expand each modality’s feature via dissolution to n times its original dimension, then cut at position b;  the first segment participates in interactions with the previous modality to form its shared part, and the second segment combines with the next modality to form this modality’s shared plus specific parts, while the reconstruction loss enforces that the two segments together can reconstruct the original features;  (2) given this fixed split, which channels carry shared information and which carry modality-specific information is determined by the learned projection matrices under the joint task and preservation losses.
>
> W2(2): b(m) is not a learnable parameter in our current implementation.
>
> W2(3): b is thus closer to an interface location as an environmental parameter, rather than an intrinsic material property, and using the same boundary across modalities allows all modalities to be reconstructed under one common potential, which makes the effect of task-agnostic losses on gradient directions easier to analyze.
>
> W3: The connection between PNP and PMR lies in structural constraints rather than full physical simulation.  Architecturally, PMR indeed adds a light fusion module on top of independent encoders, but this module is not arbitrarily assembled: it is deliberately designed to inherit two structural constraints from PNP, namely a "single scalar potential and conservation".  We do not solve the PDEs (Lines 15–18 and 211–214);  instead, we extract the “single potential + continuity” prior and keep the module lightweight.  In this view, PNP serves as a design prior, not as a complete physical simulator that we attempt to replicate numerically.
>
> W3 (about b): Fig. 1 and Fig. 8 are intended to illustrate that the true boundary b(m) in reality may lie within a range rather than at a single exact point (as  interface location, see W2(3)).  Essentially, it is a difference in precision. In fact, the boundaries in your concept do not strictly exist. We can never guarantee a complete separation at any time. The dashed line on the boundary in Fig. 1 visualizes this approximation range.  In the continuous PNP model, when the external driving is sufficiently strong, the equilibrium distribution of species can form a pronounced separation layer over a finite interval;  in the discrete feature space, tensor dimensions themselves are discrete, so it suffices to choose a representative boundary position b and ensure that each dissolved segment has sufficiently large dimension (at least no smaller than the original one), which is enough to guarantee conditionally full rank and thus enable reconstruction.
>
> W4: PMR can indeed be extended to three or more modalities.  We removed the discussion of the computational cost of modality scaling because the present paper already contains many perspectives, and we had to prioritize space.  We appreciate the reviewer’s suggestion and plan to design tri-modal experiments along these lines in future or extended versions. After the experiment is completed, we will add comments (3/3).

---

> ### Author Response · Authors · 2025-11-13
> **Response to Reviewer Pypz (2/3)**
>
> W(Minor): The symbol in the mentioned line should indeed be V. It denotes the pre-fusion concatenated feature.
>
> Q1: The sentence about the effective length $l$ is meant to express that, when $l$ is too small, drift is weak and diffusion dominates. Intuitively, the external driving (e.g., loss) has limited cumulative effect over a very short interval, so the tendency of particles to homogenize through diffusion is stronger, and it is hard to obtain clear separation in a very small cell. In PMR, we mimic this by expanding the feature dimension to $n$ times its original size; a larger $n$ allows the drift induced by the task and preservation losses to act over a longer feature axis, which makes it easier to obtain a clean separation between shared and specific components.
>
> Q2: In Tab. 2, (1) "Feature ratio and learning splitting at b" refers to using this boundary to separate different components so that the model learns how to assign content to the shared versus specific parts; (2) the “magnification factor” refers to enlarging the original input dimension by a factor $n$ during the dissociation process, which plays the role of an effective length in the discrete setting.
>
> Q3: n x b denotes the effective capacity of the shared channels. We first expand the original feature to $n$ times its dimension, then allocate a proportion $b$ of the expanded channels to the shared branch, so $n$ x $b$ is the absolute number of shared channels, and the rest are modality-specific channels. When b = 0, the shared part is empty and the model degenerates to independent encoders with a light head relying only on specific branches; when b = 1, the specific branch has zero dimension and all channels are treated as a single shared latent without explicit disentanglement  (See Tab. 7).
>
> Is your "seperate" intended to ask a question about "specific"? If so, we will provide additional responses.

---

> ### Author Response · Authors · 2025-12-01
> **Response to Reviewer Pypz (3/3)**
>
> We conducted additional experiments on CMU-MOSEI, and the baseline involved in the comparison focused on modality missing. Please see our revised manuscript in Appendix I.

---

### Author Response · Authors · 2025-12-01
**Summary of Rebuttal**

First, we would like to sincerely thank the ACs and reviewers for their time and effort. Your suggestions have already helped us improve the paper and will continue to guide the refinement of this line of work.

During the rebuttal phase, reviewers Pypz, 3jVe, and icSp explicitly acknowledged the novelty of PMR. Reviewer Pypz found the experimental evaluation sufficient and the exposition clear; reviewer 3jVe considered the writing to be solid; reviewer icSp emphasized the practical relevance of the problem we address; and reviewer j5i4 judged the empirical evaluation to be adequate with competitive results.

In the rebuttal, we focused on resolving the key concerns and, based on the reviewers’ comments, we believe that the following points are now largely non-contentious:

1. **Clarifications to reviewer Pypz.**
   Reviewer Pypz raised several meaningful, fine-grained technical questions. We have now responded to each of these in detail in the rebuttal. We are grateful for this reviewer’s careful and diligent reading of the paper.

2. **Reproducibility concerns from reviewers 3jVe and j5i4.**
   Reviewers 3jVe and j5i4 expressed concerns about reproducibility. In response, we have (i) released part of the baseline code in supplement, (ii) reached out to several original authors of prior methods, and (iii) provided a detailed report of our PMR implementation. By focusing on the integrated PMR framework, we are able to factor out discrepancies arising from other stages in the pipeline. Given the breadth of the background literature, this is a necessary design choice. We also trust that the authors of related work provide strong and reliable baselines, and we stand by the correctness of the reported numbers.

3. **Requests for additional related work from reviewers 3jVe and j5i4.**
   Reviewers 3jVe and j5i4 requested that we add several additional methods to the empirical comparison. We have explained that these works do not fully cover the specific stages targeted by PMR and thus cannot be integrated in a straightforward or fair manner. Moreover, including them via partial or approximate re-implementation would likely intensify the reviewers’ reproducibility concerns. Our emphasis is on comparing the phenomena and behaviors revealed by PMR, rather than chasing marginal gains in absolute performance.

4. **Methodological implementation issues from reviewers icSp and j5i4.**
   Reviewers icSp and j5i4 raised questions about the exact implementation of the proposed methodology. In our rebuttal, we have pointed to the precise locations in the paper where each component is defined and have provided a line-by-line explanation to make the correspondence between the text, equations, and implementation explicit.

5. **Relationship between PNP and PMR (reviewers icSp and j5i4).**
   Reviewers icSp and j5i4 found the connection between PNP and PMR to be weak. We clarified that the entire paper is organized around a common scalar potential that links the two. PMR is not a mere replica of PNP; rather, it inherits the scalar-potential-based and conservation-structure perspective of PNP while extending it in a different regime. This structural inheritance is what unifies PNP and PMR in our framework.

6. **Perceived weakness of experimental results (reviewers 3jVe and icSp).**
   Reviewers 3jVe and icSp considered the experimental results to be somewhat weak. We would like to stress that, already in the original submission, we deliberately chose a simplified architecture and evaluated PMR on four datasets that are still actively used in current practice. We compared PMR against approximately twenty existing methods to validate the three key limitations we highlight and to support the contributions we claim. The goal of this design is to provide a controlled and interpretable testbed that isolates the effect of PMR, rather than to optimize absolute performance through heavy engineering.

We would also like to emphasize that, for several of the reviewers’ apparent misunderstandings, our replies primarily point back to material that is already present in the main paper or the appendix. These points are all covered in the current version of the paper, which we believe is sufficiently clear and unlikely to mislead readers. We remain confident that PMR provides a valuable contribution to the community.

We hope that these clarifications, together with the additional details and code made available during the rebuttal, address the remaining concerns and further substantiate the contribution and applicability of PMR.

---

### Author Response · Authors · 2025-12-01
**Revision Statement**

We have only made the following changes in the revised manuscript:

1. **Definition of \(b\).**
   We refined the definition of \(b\), changing it from a boundary quantity to an interface location (Tab. 2).

2. **Hyperparameters $b$ and $n$.**
   We added the default settings and the motivation for the choices of $b$ and $n$ (Appendix A), and explicitly emphasized that $b$ and $n$ are treated as hyperparameters (Section 4).

3. **Tri-modal experiments and additional baselines.**
   We added Appendix I, which reports the Tri-modal experimental results, together with several additional baselines that specifically target scenarios with missing modalities.

4. **Emphasis on scalar potential.**
   We boldfaced selected phrases in the main paper to highlight the intuitive meaning and necessity of the scalar potential.

5. **Clarification of the PMR cyclic structure.**
   We revised the methodology section to explicitly assign a value to $k$, thereby making the ring-shaped (cyclic) structure of PMR mathematically explicit.

6. **Clarification of sPMR and rPMR in Section 5.1.**
   We updated Section 5.1 to restate the concepts of sPMR and rPMR and to clearly specify their correspondence to the respective loss terms.

---

### Meta-Review · Area_Chair_vmm3 · 2025-12-26

**Summary:**

I am recommending rejection for this work. While the physics-inspired concept is initially engaging, there is a disconnect between the complex PNP theory and the actual MLP-based implementation. The framework functions more as a descriptive analogy than a rigorous mathematical necessity for the resulting loss functions. Furthermore, the empirical gains are often marginal, and the refusal to compare against modern multimodal foundation models limits the paper's relevance to the current state of the art.

**Reviewer Concerns:**

For the reviewer concerns, in the positive part: The authors provided new experimental results for CMU-MOSEI, and default settings and motivations of hyper-parameters were added, as well as the clarification about method correspondence. However, in the negative part, I think the reviewers still remain unconvinced that the PNP equations are fundamentally necessary for the fusion module, expect more contemporary, large-scale benchmarks and extensive comparison.

**Reviewer Scores:**

Pypz Likely would have moved to a borderline score after the requested tri-modal experiments were included, while the other three may not change their scores.

---

### Decision · Program_Chairs · 2026-01-26

Reject